# Fibrinogen induces neural stem cell differentiation into astrocytes in the subventricular zone via BMP signaling

Lauriane Pous[1,2], Sachin S. Deshpande[1,2], Suvra Nath[1,2], Szilvia Mezey[1,2], Subash C. Malik[1,2], Sebastian Schildge[1,2], Christian Bohrer[1,2], Könül Topp[1,2], Dietmar Pfeifer[3], Francisco Fernández-Klett[4], Soroush Doostkam[5], Dennis K. Galanakis[6], Verdon Taylor [7], Katerina Akassoglou[8,9] & Christian Schachtrup[1,10]*

Neural stem/progenitor cells (NSPCs) originating from the subventricular zone (SVZ) contribute to brain repair during CNS disease. The microenvironment within the SVZ stem cell niche controls NSPC fate. However, extracellular factors within the niche that trigger astrogliogenesis over neurogenesis during CNS disease are unclear. Here, we show that blood-derived fibrinogen is enriched in the SVZ niche following distant cortical brain injury in mice. Fibrinogen inhibited neuronal differentiation in SVZ and hippocampal NSPCs while promoting astrogenesis via activation of the BMP receptor signaling pathway. Genetic and pharmacologic depletion of fibrinogen reduced astrocyte formation within the SVZ after cortical injury, reducing the contribution of SVZ-derived reactive astrocytes to lesion scar formation. We propose that fibrinogen is a regulator of NSPC-derived astrogenesis from the SVZ niche via BMP receptor signaling pathway following injury.

[1] Institute of Anatomy and Cell Biology, Faculty of Medicine, University of Freiburg, 79104 Freiburg, Germany. [2] Faculty of Biology, University of Freiburg, 79104 Freiburg, Germany. [3] Department of Hematology, Oncology and Stem Cell Transplantation, University Medical Center Freiburg, University of Freiburg, 79106 Freiburg, Germany. [4] Department of Neuropsychiatry & Laboratory of Molecular Psychiatry, Charité – Universitätsmedizin Berlin, 10117 Berlin, Germany. [5] Institute of Neuropathology, University Medical Center Freiburg, University of Freiburg, 79104 Freiburg, Germany. [6] Department of Pathology, State University of New York, Stony Brook, NY 11794, USA. [7] Department of Biomedicine, Embryology and Stem Cell Biology, University of Basel, Mattenstrasse 28, CH-4058 Basel, Switzerland. [8] Gladstone Institutes, San Francisco, CA 94158, USA. [9] Department of Neurology, University of California San Francisco, San Francisco, CA 94143, USA. [10] Center for Basics in NeuroModulation (NeuroModulBasics), Faculty of Medicine, University of Freiburg, Freiburg, Germany. *email: christian.schachtrup@anat.uni-freiburg.de

A fine-tuned cellular and molecular niche environment is instrumental for neural stem/precursor cell (NSPC) maintenance and directed differentiation[1–3]. Pathological states, including trauma and stroke, alter NSPC cell fate affecting tissue homeostasis and brain repair[4–6] that might be caused by dynamic changes in the NSPC niche environment. The NSPCs in the adult subventricular zone (SVZ) communicate with local cells, the vascular system and with the cerebrospinal fluid by receiving cell- or blood-derived signals that regulate proliferation and differentiation[7–9]. In the adult rodent brain SVZ, astrocyte-like type B stem cells generate type C transit amplifying cells and subsequently doublecortin-positive (DCX+) neuroblasts[10,11]. We collectively refer to B- and C-cells as NSPCs. Neuroblasts leave the SVZ and migrate long distances through the rostral migratory stream to the olfactory bulb to become interneurons[12]. Upon CNS disease, NSPCs of the adult SVZ respond with increased proliferation, a redirected migration toward the lesioned area and their differentiation primarily into astrocytes[4–6,13,14]. Understanding how central nervous system (CNS) disease alters the SVZ stem cell niche environment and the release of blood-derived factors and how these factors regulate NSPC cell fate is instrumental for harnessing these cells for brain repair.

Fibrinogen (coagulation factor I) is a 340-kDa protein secreted by hepatocytes in the liver and present in the blood circulation at 3–5 mg/ml[15,16]. Fibrinogen is cleaved by thrombin and, upon conversion to fibrin, serves as the major architectural protein component of blood clots. In CNS disease fibrinogen enters the CNS in areas with vascular permeability or blood–brain barrier (BBB) disruption and is deposited as insoluble fibrin forming a provisional extracellular matrix during brain repair[16,17]. Fibrin is present in the brain in a wide range of CNS pathologies, such as multiple sclerosis (MS), Alzheimer disease (AD), stroke, and traumatic brain injury (TBI)[16]. Fibrinogen acts as a multi-faceted signaling molecule by interacting with integrins and non-integrin receptors and by functioning as a carrier of growth factors regulating their bioavailability[16–19]. Thereby fibrinogen promotes inflammation and neurodegeneration, while it inhibits myelin repair[16]. However, the role of fibrinogen in NSPC differentiation remains unknown.

Here, we sought to determine the role of fibrinogen on NSPC fate choice in the adult SVZ niche. Using mouse models for cortical ischemic stroke (photothrombotic ischemia) and cortical brain trauma (stab wound injury), we show that fibrinogen leaks into the distant SVZ stem cell niche environment upon cortical injury. Genetic or pharmacologic depletion of fibrinogen in vivo reduced NSPC differentiation into astrocytes and lineage tracking using tamoxifen inducible *Nestin-CreER^{T2};YFP^{fl}* transgenic reporter mice in combination with pharmacologic fibrinogen depletion revealed reduced contribution of SVZ-derived Thbs4 + reactive astrocytes to lesion scar formation. Accordingly, fibrinogen inhibited neuronal differentiation of primary NSPCs from the SVZ or hippocampus and promoted their differentiation into astrocytes in vitro. Fibrinogen treatment of NSPCs induced the expression of BMP target genes, e.g. *inhibitor of DNA binding 3* (*Id3*). Exposure of NSPCs to fibrinogen deletion mutants or to integrin-blocking peptides showed that the fibrinogen αC domain containing the RGD sequence is involved in BMP signaling activation and NSPC differentiation into astrocytes. Thus, our results demonstrate that the circulating blood-derived factor fibrinogen induces an astrogenic milieu in brain stem cell niches that may determine the contribution of NSPCs in repair mechanisms in CNS disease.

## Results
**Fibrinogen deposition in the SVZ after cortical injury.** NSPC maintenance and differentiation is instructed by the extracellular

environment in their particular stem cell niche[1–3,20]. The SVZ contains an extensive planar vascular plexus with NSPCs poised to receive spatial cues and regulatory signals from diverse elements of the vascular system[7]. Since the SVZ vascular system is already permeable for circulating small molecules (sodium fluorescein, 376 Da) under homeostatic conditions[7], we examined whether blood components enter the SVZ stem cell niche environment and alter NSPC differentiation in vivo using different mouse models of CNS injury. Using the photothrombotic ischemia (PT) stroke model[21], we showed that GFAP was globally upregulated around the cortical lesion core forming the glia scar 7 days after PT, as expected (Fig. 1a). GFAP was also strongly upregulated in the SVZ (Fig. 1a, enlargement). Surprisingly, the blood-derived coagulation factor fibrinogen (340 kDa) was deposited inside the entire SVZ stem cell niche 1 day after PT (Fig. 1b, enlargement) and extravascular fibrinogen surrounded CD31+ blood vessels in the SVZ stem cell niche 5 h after stab wound injury (SWI), a mild model of cortical brain trauma (Supplementary Fig. 1a–d). Importantly, fibrinogen deposition was significantly evident in the SVZ stem cell niche after middle cerebral artery occlusion, the mouse model for stroke as well as in the SVZ of human brain sections after stroke (Fig. 1c, d, Supplementary Fig. 1e), suggesting that fibrinogen deposition in the SVZ stem cell niche environment is a general feature in stroke. Fibrinogen was not significantly detected in the hippocampal or third ventricle stem cell niche after PT (Supplementary Fig. 2a, b).

Next, we examined the potential route of entry of the circulating blood factor fibrinogen into the SVZ. SVZ NSPC behavior is affected by receiving cues from the lateral ventricle-derived cerebrospinal fluid and from the SVZ vascular system[7,22]. Fibrinogen deposits were not observed in the S100β + ependymal cell layer lining the lateral ventricle (Fig. 1e), suggesting that the cerebrospinal fluid was not the source of fibrinogen. Fibrinogen surrounded CD31 + blood vessels in the SVZ (Fig. 1f, enlargements on the right, bottom, Supplementary Fig. 1d), suggesting leakage of blood vessels in the SVZ. By contrast, fibrinogen was not found surrounding CD31 + blood vessels in the striatum implying an intact BBB (Fig. 1f, enlargements on the right, top). The leaky BBB unique to the SVZ even under homeostatic conditions is caused by a reduced pericyte coverage of the vasculature[7]. Fibrinogen has been shown to induce cell death of pericytes[23], potentially resulting in the increased leakage of the SVZ BBB in CNS disease. However, pericyte cell numbers and vascular coverage were not affected 1 day after PT (Supplementary Fig. 2c), suggesting that fibrinogen deposition in the SVZ environment was not a result of increased pericyte cell death. Detailed analysis showed that fibrinogen immunoreactivity increased rapidly in the SVZ niche 1 day after PT and colocalized with significantly increased GFAP + cell numbers in the ipsilateral and contralateral SVZ after PT (Fig. 1g, Supplementary Fig. 2d). While fibrinogen deposition in the SVZ stem cell niche drastically decreased on day 3 after PT, GFAP + cells gradually decreased 7 and 14 days after PT (Fig. 1g, Supplementary Fig. 2d). These results showed that a cortical brain insult leads to fibrinogen deposition into the SVZ, potentially by increased leakiness of the specialized SVZ vasculature.

**Fibrinogen induces NSPC differentiation into astrocytes.** CNS disease triggers SVZ-derived NSPC migration towards the lesion area, where they mainly differentiate into glial cells[4–6]. As fibrinogen was promptly deposited in the SVZ niche environment after cortical brain injury and since fibrinogen colocalized with Nestin + stem cells after PT (Supplementary Fig. 3a), we sought to determine the direct role of fibrinogen on adult NSPC differentiation in vitro. Adult SVZ-derived NSPCs are multipotent

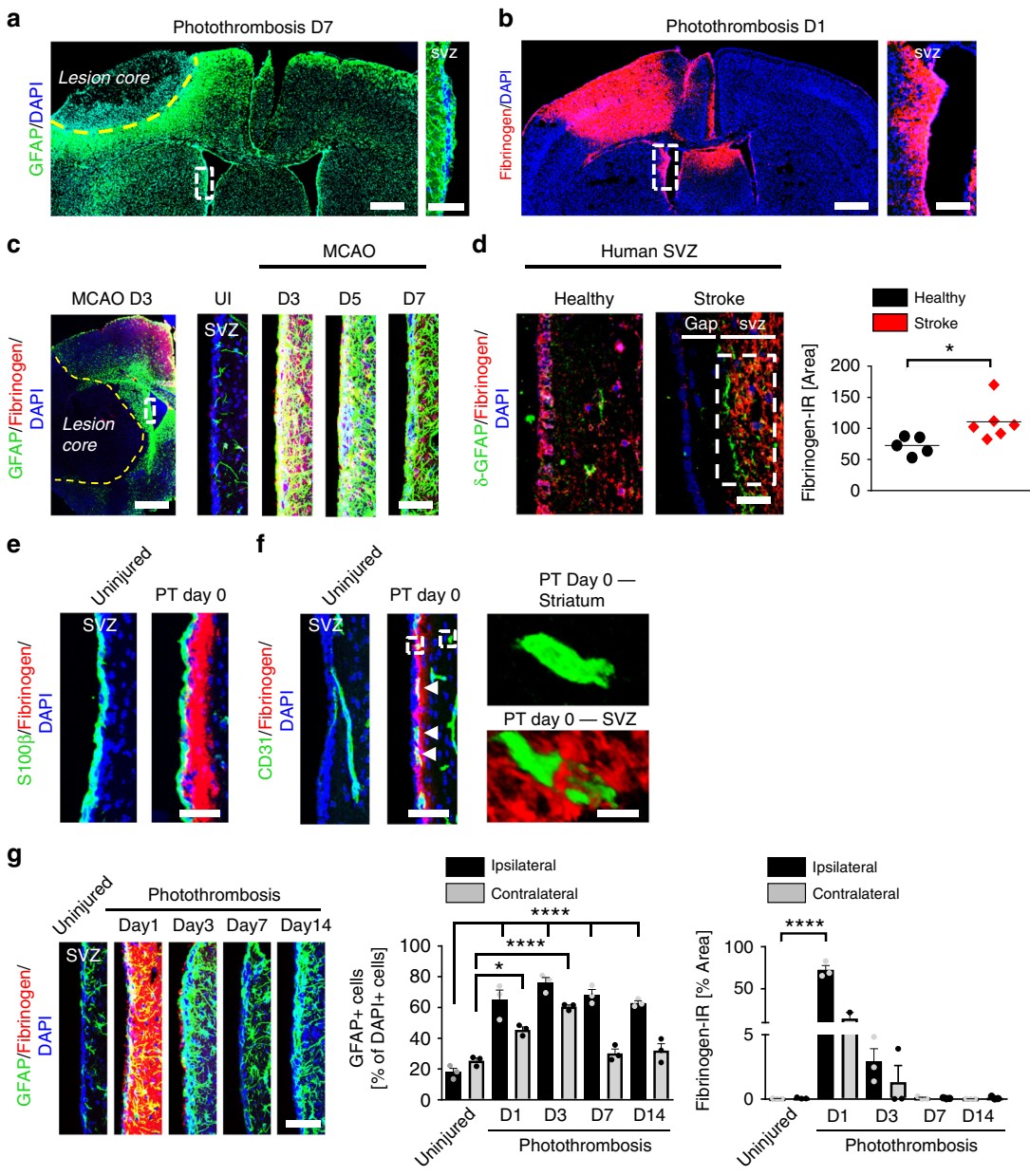

**Fig. 1 Fibrinogen deposition in the SVZ after cortical injury. a** GFAP immunostaining (green) around the lesion core (indicated by yellow dotted line) and in the SVZ. White box indicating enlargement of SVZ showing GFAP expression in the adult SVZ 7 days after PT (right) (n = 4 mice). Scale bars, 369 μm, left and 57 μm, enlargement. **b** Fibrinogen immunostaining (red) in the lesion core and in the SVZ 1 day after PT. White box indicating enlargement of SVZ showing fibrinogen deposition in the SVZ 1 day after PT (right) (n = 4 mice). Scale bars, 428 μm, left and 120 μm, enlargement. **c** GFAP (green) and fibrinogen (red) immunostainings in the lesion core (left) and in the SVZ (right) 3, 5 and 7 days after stroke (MCAO). White box indicating enlargement of SVZ showing increased GFAP expression colocalizing with fibrinogen (red) deposition in the SVZ 3 days after stroke. Scale bars, 543 μm, left and 50 μm, right (n = 3 mice). **d** δ-GFAP (green) and fibrinogen (red) immunostainings in the SVZ of patients after stroke compared to healthy controls. White box indicating quantification area. Scale bar, 25 μm. Quantification of fibrinogen immunoreactivity in the SVZ per area (n = 5 (healthy control), n = 6 (stroke patients, mean ± s.e.m., unpaired Student's t-test, *P < 0.05). **e** Representative images of S100β + ependymal cells (green) lining the lateral ventricle and adjacent fibrinogen (red) in the SVZ 5 h after PT (PT Day 0). Scale bar, 40 μm. **f** CD31 + endothelial cells (green) and fibrinogen (red) in the SVZ and striatum 5 h after PT (PT Day 0). White boxes indicating enlargement of fibrinogen deposition around blood vessels in SVZ (right, bottom) compared to no fibrinogen deposition around blood vessels in the striatum (right, top) after PT. Scale bars, 40 μm, left and 95 μm, enlargement. **g** GFAP + astrocytes (green) and fibrinogen (red) in the ipsilateral SVZ at different days after PT and uninjured mice. Scale bar, 57 μm. Quantification of GFAP + cells and fibrinogen immunoreactivity in the SVZ per area (n = 3 mice, mean ± s.e.m., one-way ANOVA and Bonferroni's multiple comparisons test, *P < 0.05, ****P < 0.0001). SVZ subventricular zone.

stem cells in culture (Supplementary Fig. 3b) with the intrinsic ability to form neurospheres and to differentiate into neurons, astrocytes and oligodendrocytes upon growth factor withdrawal and adherent culture conditions[24]. Extracellular signals drive NSPCs toward a glial or a neuronal fate[25]. Fibrinogen induced

adult SVZ-derived and hippocampal-derived NSPC differentiation into astrocytes, revealed by increased abundance of GFAP+, Aldh1l1+, and Aqp4+ cells and an increase in *GFAP*, *Aqp4*, and *Aldoc* mRNA and protein expressed by astrocytes (Fig. 2a–e; Supplementary Fig. 3c). Fibrinogen treatment of SVZ- and

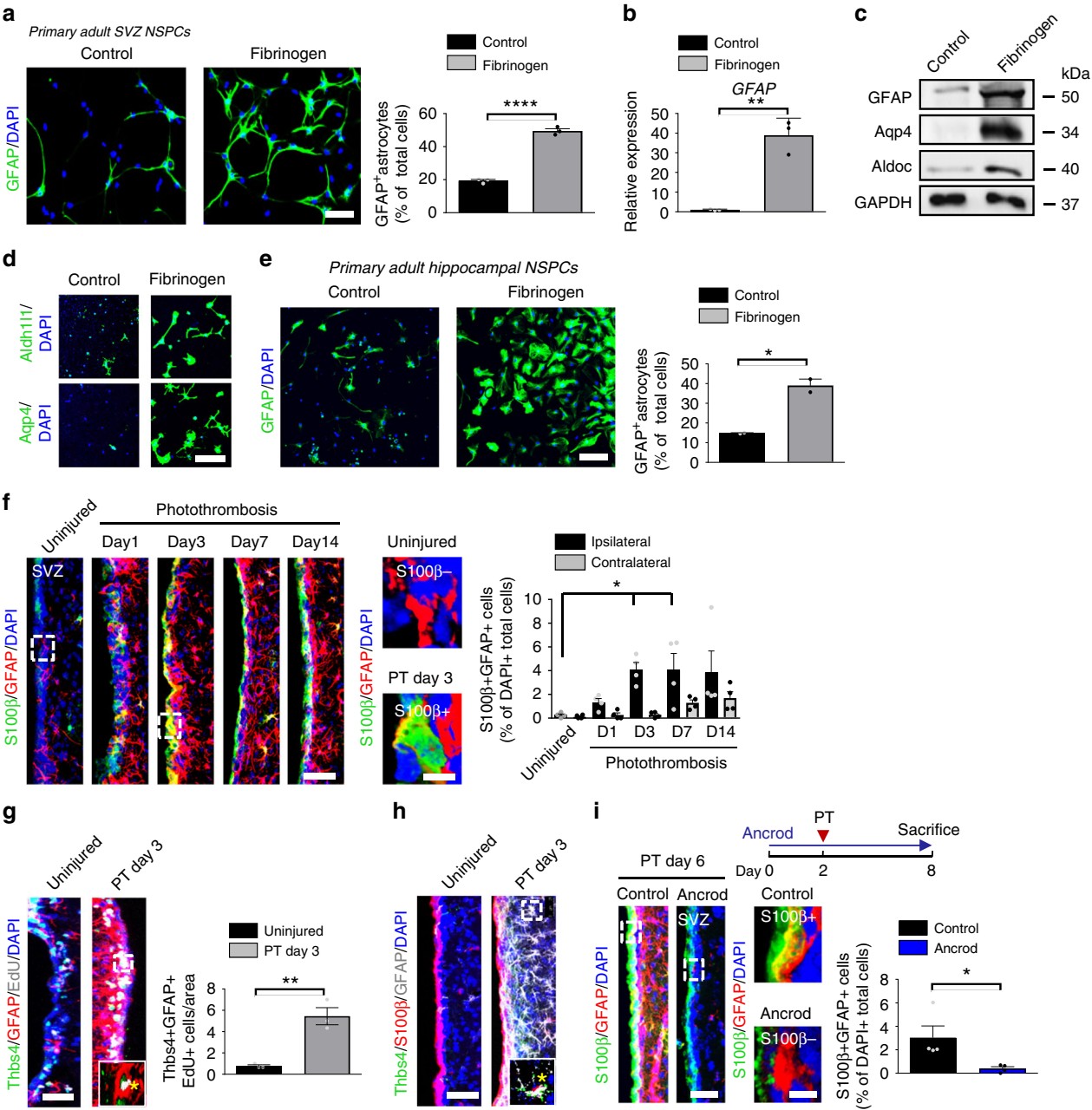

**Fig. 2 Fibrinogen-induced differentiation of NSPCs into astrocytes. a** GFAP + astrocytes (green) in untreated and fibrinogen-treated adult SVZ-derived NSPCs. Scale bar, 56 μm. Quantification of GFAP + astrocytes. ($n = 3$, mean ± s.e.m, unpaired Student's $t$-test, ***$P < 0.001$). **b** Expression of *GFAP* mRNA in NSPCs. ($n = 3$, mean ± s.e.m, unpaired Student's $t$-test, **$P < 0.01$). **c** Immunoblot for GFAP, Aqp4, Aldoc, and GAPDH in NSPCs 2 days after fibrinogen treatment. **d** Aldh1l1 + (green, top) and Aqp4 + (green, bottom) astrocytes in fibrinogen-treated adult SVZ-derived NSPC cultures. ($n = 3$, representative images are shown). Scale bar, 67 μm. **e** GFAP + astrocytes (green) in fibrinogen-treated hippocampus-derived NSPC cultures. Scale bar, 104 μm. Quantification of GFAP + astrocytes. ($n = 2$, performed in duplicate, mean ± s.e.m, unpaired Student's $t$-test, *$P < 0.05$. **f** S100β (green) and GFAP (red) immunostainings in the SVZ at different timepoints after PT. White boxes indicate enlargement of S100β-GFAP + (right, top) and S100β + GFAP + (right, bottom) cells in the SVZ of uninjured and injured (PT day 3) mice, respectively. Scale bars, 40 μm, left and 7 μm, enlargement. Quantification of S100β + GFAP + cells per area ($n = 4$ mice, mean ± s.e.m, one-way ANOVA and Bonferroni's multiple comparisons test, *$P < 0.05$). **g** EdU (grey), Thbs4 (green) and GFAP (red) immunostainings in the SVZ at 3 days after PT. White box indicates the enlargement of an EdU + GFAP + Thbs4 + DAPI + cell (right, bottom) in the SVZ 3 days after PT. Scale bar, 25 μm. Quantification of Thbs4 + GFAP + EdU + cells in the SVZ per area. ($n = 4$ mice, mean ± s.e.m, unpaired Student's $t$-test, **$P < 0.01$). **h** immunolabeling for Thbs4 (green), S100β (red) and GFAP (gray) in the SVZ at 3 days after PT. White box indicating enlargement of a Thbs4 + S100β + GFAP + DAPI + cell in the SVZ 3 days after PT ($n = 4$ mice). Scale bar, 25 μm. **i** Scheme illustrating PT on ancrod-administered WT mice (top). S100β (red) and GFAP (green) immunostainings of the SVZ of ancrod-treated mice 6 days after PT. White boxes indicate enlargement of a S100β + GFAP + (top) and a S100β-GFAP + (bottom) cell in the SVZ of control and fibrinogen-depleted mice, respectively, 6 days after PT. Scale bars, 36 μm, left and 7 μm, enlargement. Quantification of S100β + GFAP + cells in the SVZ per area. ($n = 4$ mice, unpaired Student's $t$-test, *$P < 0.05$). SVZ, subventricular zone.

hippocampal-derived NSPCs decreased the fraction of Tuj-1+ neurons by 61% and 95%, respectively (Supplementary Fig. 3d, e). In contrast to the treatment of hippocampal-derived NSPCs, fibrinogen treatment of SVZ NSPCs increased the cell number and decreased apoptosis (Supplementary Fig. 3f, g). Overall, these data suggest that fibrinogen induced the differentiation of adult NSPCs into astrocytes.

**Fibrinogen deficiency reduces SVZ astrocyte formation**. To investigate the role of fibrinogen on adult NSPC fate in vivo, we first examined whether the changed SVZ niche environment after cortical brain injury already induced NSPC differentiation into astrocytes within the SVZ. Therefore, we examined the expression of the astrocyte marker GFAP in combination with S100β, a marker for mature astrocytes that have lost their neural stem cell potential[26], at different timepoints after PT. GFAP + S100β+ astrocytes were upregulated by 16.5-fold in the ipsilateral SVZ 3 days after PT compared to uninjured mice (Fig. 2f). In accordance, using GFAP in combination with Aqp4, another marker upregulated in mature astrocytes[27], revealed a 5.7-fold increase of GFAP + Aqp4+ astrocytes in the ipsilateral SVZ 3 days after PT (Supplementary Fig. 4a). Interestingly, SVZ astrocytes did not express Lcn2, a marker of reactive astrocytes[28], 3 days after PT (Supplementary Fig. 4b), suggesting that fibrinogen deposition in the SVZ increased the number of astrocytes without inducing activation. Next, we thought to identify the source of the increased number of newborn, mature astrocytes in the SVZ after cortical PT. Proliferation of mature astrocytes[29] and/or proliferation of NSPC subpopulations and their differentiation towards astrocytes could be potential sources. Immunolabeling of EdU + GFAP + S100β + cells failed to detect any proliferating mature astrocytes in the SVZ after PT (Supplementary Fig. 4c); however, we detected EdU + cells co-labeled with the NSPC and astrocyte marker GFAP+ in the SVZ (Supplementary Fig. 4c, enlargement top). Detailed analysis of SVZ subpopulations revealed that Thbs4 + NSPCs significantly increased until day 3 after PT and thereafter dropped down to basal level, while the total cell number and the number of apoptotic cells did not change significantly at any of the investigated timepoints in the SVZ after PT (Supplementary Fig. 4d). Characterization of the Thbs4 + cell population revealed a 5.7-fold increase in their proliferation (EdU + Thbs4 + GFAP + ) 3 days after PT (Fig. 2g) and their further differentiation into mature astrocytes (Thbs4 + GFAP + S100β + ; Fig. 2h). Loss of fibrinogen by acute fibrinogen depletion using the pharmacologic reagent ancrod or by using the fibrinogen-deficient (Fgα−/−) mouse line resulted in a 87% and 74% reduction of GFAP + S100β + astrocytes in the SVZ at 6 and 3 days post-injury compared to control mice, respectively (Fig. 2i, Supplementary Fig. 4e). Neither uninjured Fgα−/− mice nor ancrod-treated animals showed significant differences in the NSPC population compared to controls (Supplementary Fig. 5a–c). Overall, these results suggest that fibrinogen deposition in the SVZ environment induces NSPC differentiation into astrocytes after cortical brain injury.

**Fibrinogen induces astrogliogenesis via the BMP–Id3 axis**. To identify the molecular mechanisms fibrinogen utilizes to induce the differentiation of NSPCs into astrocytes, we compared the gene expression profile of cultured WT NSPCs 12 h after fibrinogen treatment to untreated cells by microarray analysis. Applying a significance threshold of 4-fold up or downregulation with a q-value of 0.005 resulted in 169 differentially regulated genes (Fig. 3a). Upon fibrinogen treatment, adult NSPCs showed an increased expression of genes known to be upregulated by reactive astrocytes upon brain injury, including GFAP and Aqp4

(Supplementary Table 1). Interestingly, adult NSPCs showed an increased expression of the neuron-survival promoting chondroitin/dermatan sulfate proteoglycan biglycan[30], but not of neuron-inhibitory scar forming genes of the chondroitin sulfate proteoglycan family (CSPGs), known to be upregulated by reactive astrocytes upon brain injury (Fig. 3a and Supplementary Table 1). Biglycan was upregulated in the SVZ 10 days after cortical injury (Supplementary Fig. 6a, left and enlargement bottom) and in BrdU + GFAP + SVZ-derived newborn astrocytes in the cortical lesion area (Supplementary Fig. 6a, right and enlargement bottom). In accordance, biglycan was expressed and secreted by fibrinogen-treated primary NSPCs, but not by fibrinogen-treated cortical astrocytes (Supplementary Fig. 6b), suggesting different functionalities of fibrinogen-induced NSPC-derived newborn astrocytes compared to cortical resident astrocytes in the lesion area. The majority of the differentially expressed genes were downregulated in NSPCs upon fibrinogen treatment compared to untreated NSPCs, including cell cycle regulators (Cyclin B1, E2f8), and transcription factors promoting neurogenesis and oligodendrogenesis (Neurod4, Myt1) (Fig. 3a and Supplementary Table 1). Furthermore, genes associated with the bone morphogenetic protein (BMP) signaling pathway, including Smad6, Smad9, and Id3, were upregulated in fibrinogen-treated NSPCs, suggesting a role for BMP signaling in the fibrinogen-induced NSPC differentiation into astrocytes (Fig. 3a and Supplementary Table 1). Using quantitative PCR we confirmed that fibrinogen treatment of adult NSPCs resulted in reduced expression of the cell cycle genes Cyclin B1 and E2f8 and increased expression of BMP-responsive genes Id3 and Smad6 (Supplementary Fig. 7a). In primary NSPCs from the SVZ and hippocampus fibrinogen induced Smad1/5/8 phosphorylation (P-Smad1/5/8), the transcriptional mediators of the BMP signaling pathway (Fig. 3b, Supplementary 7b, c). The selective inhibitor of BMP type I receptor kinases, LDN-193189[31], inhibited the fibrinogen-induced phosphorylation of Smad1/5/8 (Fig. 3c), and significantly reduced the fibrinogen-mediated adult NSPC differentiation into astrocytes (Fig. 3d), indicating that fibrinogen triggered activation of the BMP type I receptor pathway is necessary to induce NSPC differentiation into astrocytes.

We next sought to identify the downstream transcriptional mediator of fibrinogen triggered NSPC differentiation into astrocytes. We focused on Id3, which was identified in our gene array as a fibrinogen-induced target gene and which promotes adult NSPC differentiation into astrocytes[5]. While fibrinogen treatment of WT NSPCs induced a robust increase in the appearance of GFAP + astrocytes 4 days after initiation of differentiation, Id3−/− NSPCs were resistant to fibrinogen-induced differentiation into astrocytes in vitro (Fig. 3e). Id3 was rapidly upregulated in YFP+ stem cells in the SVZ 1 day after PT, while pharmacologic depletion of fibrinogen reduced Id3 expression in YFP+ stem cells in the SVZ compared to control animals after PT (Fig. 3f), suggesting that fibrinogen deposition in the SVZ stem cell environment after cortical brain injury rapidly upregulates Id3, which in turn promotes NSPC differentiation into astrocytes.

**Fibrinogen alphaC domain induces astrogliogenesis**. Fibrinogen acts as a multi-faceted signaling molecule mediated by binding to integrin cell receptors and by carriage of growth factors and cytokines[17,19]. Blocking free BMP with noggin, or with the BMP-9 scavenger endoglin showed no effect on fibrinogen-mediated Smad phosphorylation and adult NSPC differentiation into astrocytes (Fig. 4a, Supplementary 7d-e), suggesting that fibrinogen was not a carrier of free BMP. Fibrinogen contains multiple non-overlapping binding motifs for different receptors, including the tripeptide

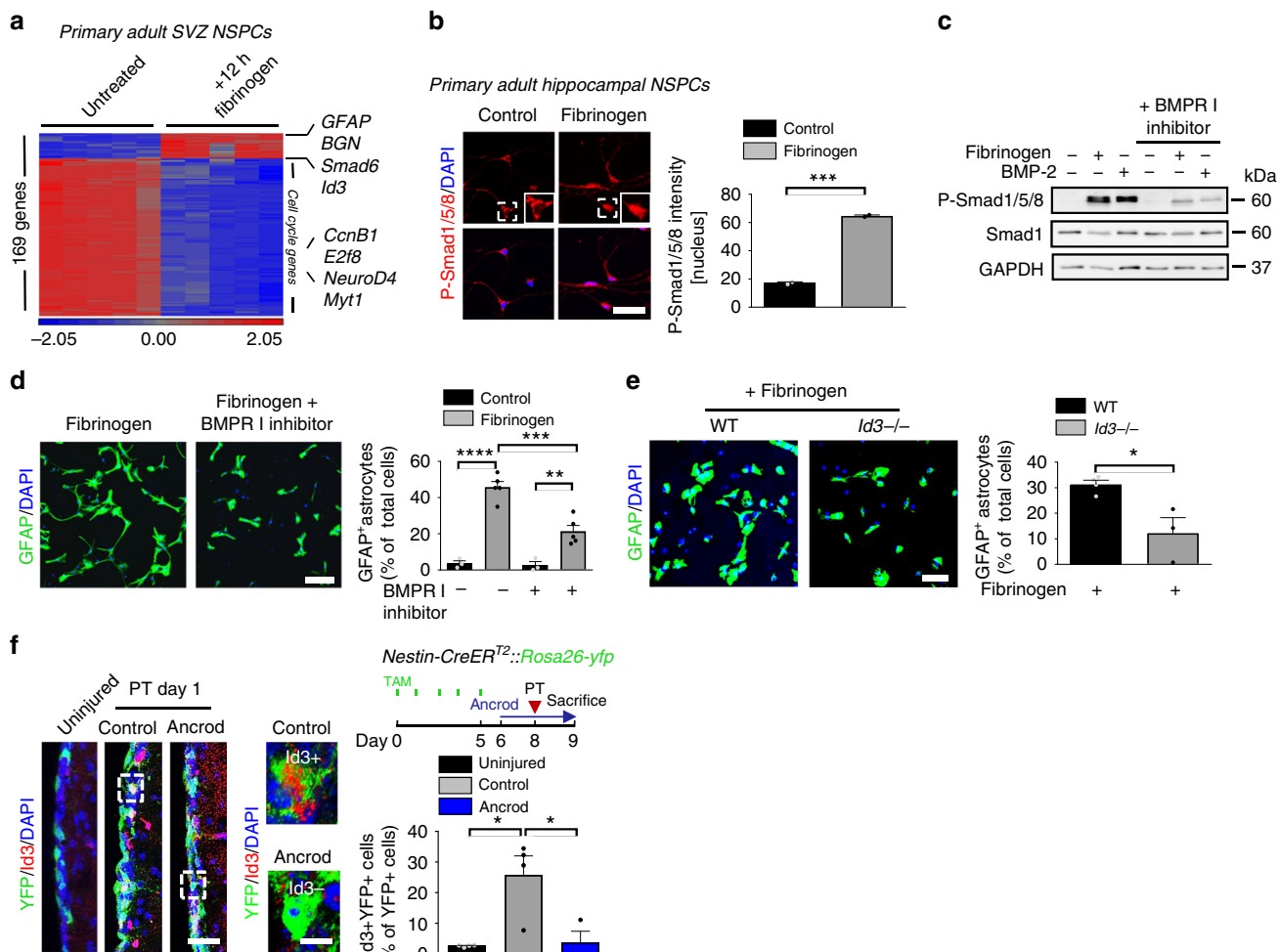

**Fig. 3 Fibrinogen induces astrogliogenesis via the BMP–Id3 axis. a** Microarray gene expression profile of NSPCs treated for 12 h with fibrinogen compared to control cells. Heatmap analysis showing genes regulated by a factor of at least 4 between fibrinogen-treated and control NSPCs. ($n = 3$). **b** Immunolabeling for phosphorylated Smad1/5/8 (P-Smad1/5/8, red) in fibrinogen-treated hippocampal NSPCs. White boxes indicate enlargement of nuclear P-Smad1/5/8- and P-Smad1/5/8 + cell in untreated and fibrinogen-treated conditions, respectively. Scale bar, 46 µm. Quantification of nuclear P-Smad1/5/8. ($n = 2$, experiments performed in duplicates (mean ± s.e.m., unpaired Student's $t$-test, ***$P < 0.001$). **c** Immunoblot for P-Smad1/5/8 and Smad1 in NSPCs pretreated with a BMP receptor inhibitor before fibrinogen stimulation. BMP-2 served as positive control. **d** GFAP + astrocytes in NSPC cultures pretreated with a BMP receptor inhibitor 1 h before fibrinogen treatment 2 days after initiation of differentiation. Scale bar, 125 µm. Quantification of GFAP + astrocytes. ($n = 5$, (mean ± s.e.m, one-way ANOVA and Bonferroni's multiple comparisons test, **$P < 0.01$, ***$P < 0.001$, ****$P < 0.0001$). **e** GFAP + astrocytes (green) in fibrinogen-treated $Id3^{-/-}$ and WT NSPCs cultures after 2 days on poly-D-lysine. Scale bar, 72 µm. Quantification of GFAP + astrocytes. ($n = 4$, WT; $n = 3$, $Id3^{-/-}$ cells, mean ± s.e.m, unpaired Student's $t$-test, *$P < 0.05$). **f** Experimental setup for photothrombosis on *Nestin-CreER^{T2}::Rosa26-yfp* mice. TAM: tamoxifen (right, top). Id3 (red) and YFP (green) immunostainings in the SVZ of uninjured mice and of ancrod-treated mice compared to control WT mice 1 day after PT. The white boxes indicate the enlargement of an Id3 + YFP + (right, top) and an Id3-YFP + (right, bottom) cell in the SVZ of control mice and fibrinogen-depleted mice, respectively, 1 day after PT. Scale bars, 30 µm, left and 8 µm, enlargement. Quantification of Id3 + YFP + cells in the SVZ per area. ($n = 4$ uninjured mice, $n = 4$ control mice after PT, $n = 3$ ancrod mice after PT, unpaired Student's $t$-test, *$P < 0.05$).

Arg-Gly–Asp (RGD) sequence on its alpha chain, which serves as a binding site for integrins[17,32]. β1-Integrin alters stem cell BMP type I receptor localization and attenuates astrogliosis[33]. To examine whether the fibrinogen-integrin interaction was necessary for the induction of BMP type I receptor activation and NSPC differentiation into astrocytes, we treated primary NSPCs with a plasma-derived fibrinogen fraction that lacks most of the fibrinogen alphaC chain including the RGD integrin recognition site (termed Des-αC-fibrinogen)[34] (Fig. 4b). Treatment of primary NSPCs with the Des-αC-fibrinogen fraction failed to induce Smad1/5/8 phosphorylation in primary NSPCs (Fig. 4c) and NSPC differentiation into astrocytes (Fig. 4d), suggesting that fibrinogen facilitates BMP receptor activation via its integrin interacting αC domain inducing NSPC differentiation into astrocytes. In accordance, fibrinogen colocalized

with active β1-integrin on SVZ-derived NSPCs (Fig. 4e) and pre-treating NSPCs with an integrin-blocking peptide reduced the fibrinogen-mediated NSPC differentiation into astrocytes (Fig. 4f). Finally, pretreatment of NSPCs with the lipid raft disrupting methyl-β-cyclodextrin reduced fibrinogen-induced P-Smad1/5/8 levels by 57% (Supplementary Fig. 7f), suggesting that fibrinogen via its integrin interacting αC domain induces BMP type I receptor localization to lipid rafts to activate BMP signaling.

**Fibrinogen-induced SVZ astrocytes add to cortical scar.** Next, we investigated whether fibrinogen-induced SVZ-generated Thbs4+ astrocytes migrated to the cortical lesion area and contributed to the glia scar after PT. Fibrinogen depletion by ancrod

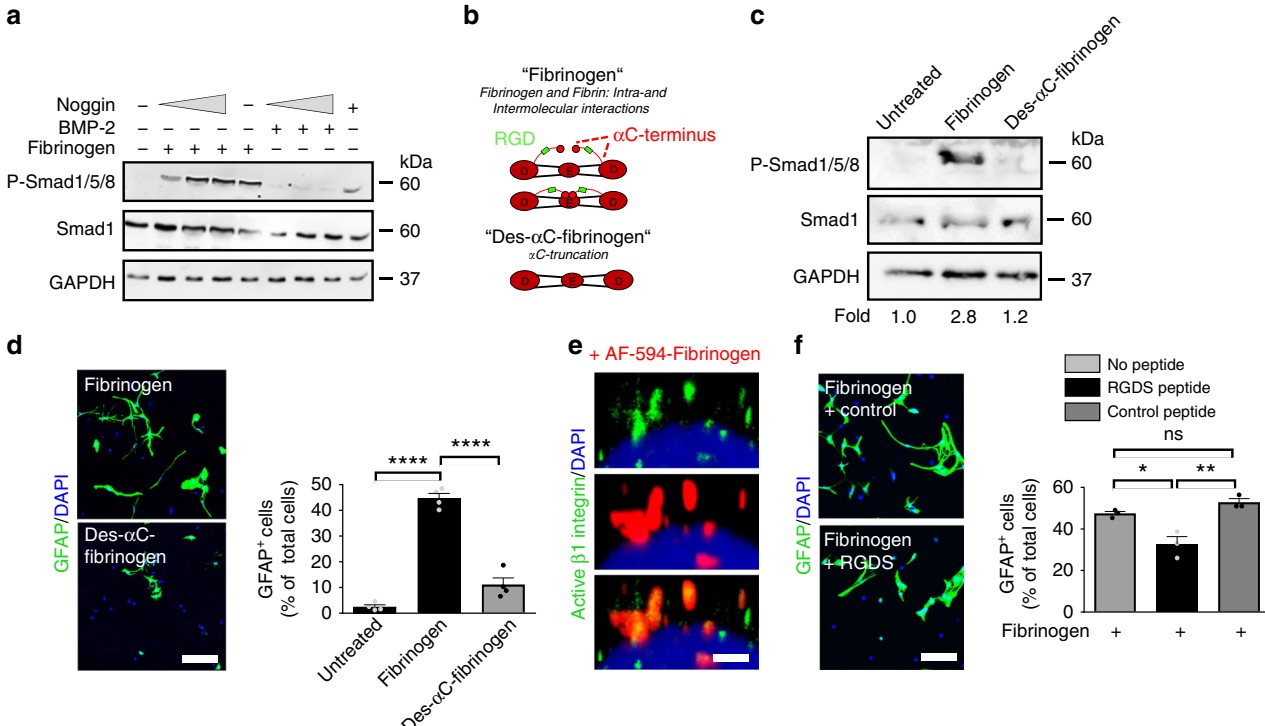

**Fig. 4 Fibrinogen αC domain requirement for NSPC differentiation into astrocytes. a** Immunoblot for P-Smad1/5/8 and Smad1 in NSPCs pretreated with an increasing concentration of Noggin before fibrinogen stimulation. BMP-2 served as positive control. **b** Scheme illustrating fibrinogen fractions isolated from plasma. Human Des-αC-fibrinogen lacks most of the residues of the C-terminus of the Aα chain including the RGD sequence. **c** Immunoblot for P-Smad1/5/8 and total Smad1 protein expression in NSPCs 2 days after fibrinogen or Des-αC-fibrinogen treatment. Values are mean of $n = 3$ independent experiments. **d** GFAP + astrocytes in fibrinogen- and Des-αC-fibrinogen-treated NSPC cultures 2 days after initiation of differentiation. Scale bar, 125 μm. Quantification of GFAP + astrocytes derived from fibrinogen- or Des-αC-fibrinogen-treated NSPCs. Results are from four independent experiments (mean ± s.e.m, one-way ANOVA and Bonferroni's multiple comparisons test, ****$P < 0.0001$). **e** Active β1-integrin (green) immunostaining in WT NSPCs treated for 15 min with AF594-fibrinogen (red). Scale bar, 2 μm. Representative images of two independent experiments are shown. **f** GFAP + astrocytes in untreated and fibrinogen-treated NSPC cultures pretreated with the integrin-blocking RGDS or the control PDEA peptides for 2 days. Scale bar, 60 μm. Quantification of GFAP + astrocytes derived from untreated and fibrinogen-treated NSPCs pretreated with the PDEA (control) and RGDS peptides. Results from three independent experiments performed in duplicates are shown (mean ± s.e.m, one-way ANOVA and Bonferroni's multiple comparisons test, *$P < 0.05$, **$P < 0.01$, ns not significant).

reduced GFAP and CSPG expression levels in the penumbra by 75% and 72%, respectively, confirming a robust effect of fibrinogen on the activation status of the astrocyte population forming the glial scar in the lesion area[18], including local and SVZ-derived newborn astrocytes[4–6,29,35] (Fig. 5a, b). Interestingly, fibrinogen depletion drastically reduced the number of Thbs4 + GFAP + astrocytes in the penumbra by 84% compared to control mice 6 days after PT (Fig. 5c). As our data suggested that fibrinogen regulates the differentiation of the SVZ Thbs4 + NSPC subpopulation into mature astrocytes within the SVZ (Thbs4 + GFAP + S100β +, Fig. 2), it is likely that these cells contribute to the glial scar in the lesion area.

Finally, we investigated the role of fibrinogen on the cell fate of SVZ-derived NSPCs by cell tracking using tamoxifen inducible *Nestin-CreER^{T2};YFP^{fl}* reporter transgenic mice[36] or by using 5-bromo-2'-deoxyuridine (BrdU) labeling regime[5]. Remarkably, immunolabeling of SVZ-derived YFP+ cells in combination with Thbs4 and GFAP in the lesion area revealed a 64.5% reduction of SVZ-derived astrocytes in fibrinogen-depleted animals compared to control animals 10 days after PT (Fig. 5d), while the overall number of SVZ-derived YFP + Thbs4 + cells at the lesion scar was not changed. The SVZ-derived YFP + Thbs4 + cells were able to penetrate the astrocyte scar border and migrate into the lesion core in fibrinogen-depleted animals (Fig. 5d, white arrowheads). In accordance, elimination of fibrinogen by using

the *Fgα^{−/−}* mouse line resulted in a 50% reduction in the number of SVZ-derived BrdU+GFAP+ reactive astrocytes in the lesion area 10 days after SWI compared to control mice (Supplementary Fig. 9a–c) and fibrinogen was sufficient to induce adult SVZ NSPC differentiation into BrdU+GFAP+ astrocytes by stereotactically injecting fibrinogen into the cortex in close proximity to the SVZ (Supplementary Fig. 9d, e). Importantly, fibrinogen-depleted mice showed an increased number of SVZ-originated YFP+ cells that migrated to the glomerular layer of the OB and an increased number of SVZ-derived cells that differentiated into YFP+ NeuN+ neurons compared to control treated animals 10 days after PT (Supplementary Fig. 10), further substantiating our findings that fibrinogen deposition within the SVZ after PT drives NSPC differentiation into astrocytes at the expense of olfactory bulb neurogenesis. Overall, these results reveal that fibrinogen drives the SVZ-derived reactive astrocyte contribution to the cortical scar formation after brain injury.

## Discussion
The identity of extracellular environmental factors regulating the differentiation of adult NSPCs into astrocytes in CNS disease is a fundamental and unresolved question. Our results suggest that upon CNS injury or disease, the SVZ vasculature becomes leaky for circulating blood proteins that dramatically change the SVZ stem cell niche environment. The blood-derived coagulation

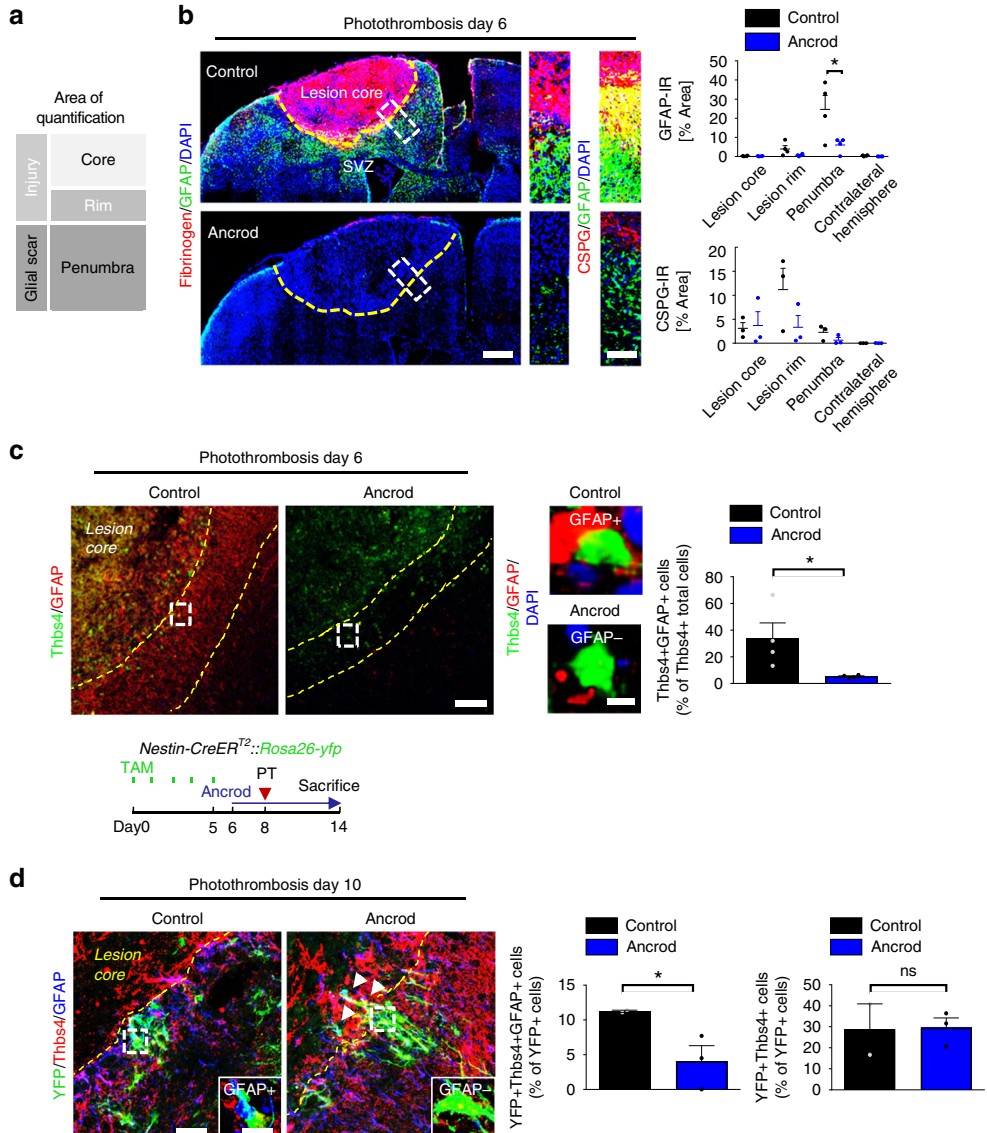

**Fig. 5 Fibrinogen-induced SVZ astrocytes add to cortical scar. a** Representation of the different lesion regions quantified by immunoreactivity. **b** Fibrinogen deposition (red) and GFAP + astrocytes in the brain of fibrinogen-depleted mice (Ancrod) compared to controls (NaCl) at 6 days after PT (left). Yellow dotted line delineates the lesion area. The white boxes indicate the enlargement of the lesion region quantified for GFAP (right, top) and CSPGs (right, bottom) immunoreactivities. Quantifications of GFAP and CSPG immunoreactivities in the different lesion regions ($n = 4$ mice, mean ± s.e.m, one-way ANOVA and Bonferroni's multiple comparisons test, *$P < 0.05$). **c** Immunolabeling for Thbs4 (green) and GFAP (red) in the lesion area of ancrod and control mice at 6 days after PT. Yellow dotted lines delineate the penumbra. The white boxes indicate the enlargement of a Thbs4 + GFAP + cell (top, right) in controls and Thbs4 + GFAP- cell in ancrod mice (right, bottom). Quantification of Thbs4 + GFAP + astrocytes in the penumbra of ancrod and control mice ($n = 4$ mice, mean ± s.e.m, unpaired Student's t-test, *$P < 0,05$). **d** Experimetal setup for ancrod-administered *Nestin-CreER^{T2}::Rosa26-yfp* mice. TAM: tamoxifen (top). YFP (green), Thbs4 (red) and GFAP (blue) immunostainings in the lesion area of ancrod and control mice at 10 days after PT. Yellow dotted lines delineate the lesion area. The white boxes indicate the enlargement of an YFP + Thbs4 + GFAP + cell in control and YFP + Thbs4 + GFAP- cell in ancrod mice. Quantifications of YFP + Thbs4 + GFAP + astrocytes (left) and YFP + Thbs4 + cells (right) in the penumbra of ancrod and control mice ($n = 2$ control mice, $n = 3$ ancrod mice, unpaired, one-tailed Student's t-test, *$P < 0,05$, ns, not significant). SVZ subventricular zone.

factor fibrinogen drives the differentiation of NSPCs into astrocytes via a BMP signaling-dependent mechanism. In the uninjured CNS, SVZ-derived adult NSPCs migrate along the rostral migratory stream to the olfactory bulbs to differentiate into interneurons. Our results suggest the following working model in CNS disease: (i) cortical brain injury increases SVZ vasculature permeability and allows fibrinogen deposition in the microenvironment of SVZ stem cells (Fig. 6). (ii) fibrinogen activates BMP type I receptor signaling in NSPCs through its αC domain. (iii) fibrinogen induces Smad1 phosphorylation and increases the expression of the transcriptional regulator Id3 through the BMP

type I receptor, thereby promoting Thbs4 + NSPC differentiation into potentially neuroprotective astrocytes (Fig. 6). Thus, local fibrinogen deposition might exert a critical role in regulating the fate of different stem cell types with neurogenic potential at sites of vascular extravasation.

Our study suggests that the 340 kDa protein fibrinogen rapidly leaks from the vasculature into the ipsilateral and contralateral SVZ upon a distant cortical brain injury. Vascular-derived small molecules have direct access to the SVZ under homeostatic conditions and SVZ NSPCs perceive small circulating blood molecules to maintain continuous neurogenesis[7]. The SVZ

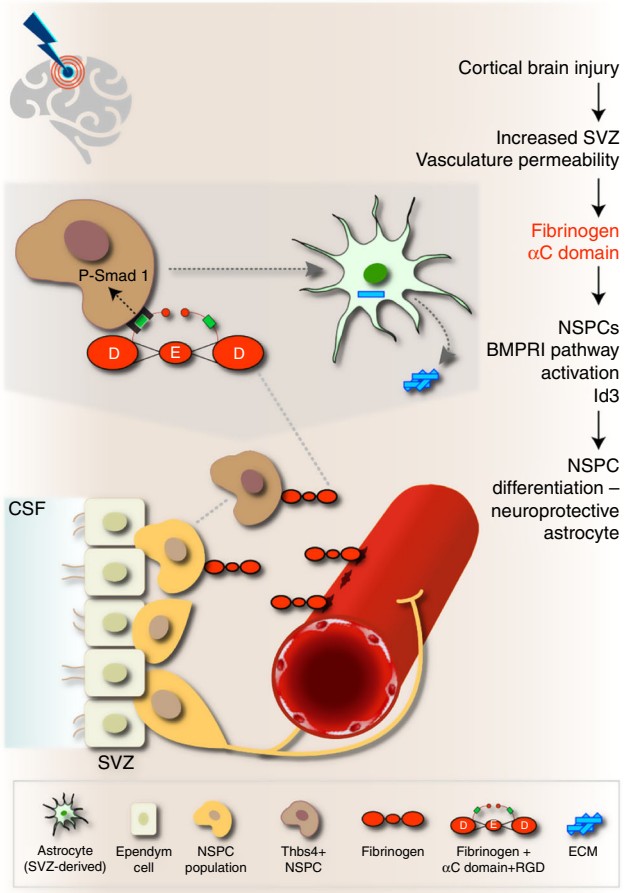

**Fig. 6 Working model for the role of fibrinogen on SVZ NSPCs in CNS disease.** In the healthy brain, NSPCs of the SVZ continuously generate mobile DCX+ neuroblasts that migrate through the rostral migratory stream to the olfactory bulb to become newborn neurons. Cortical injury results in increased SVZ vasculature permeability and fibrinogen deposition into the SVZ stem cell niche environment. Fibrinogen activates BMP receptor signaling via its αC domain inducing NSPC differentiation into neuroprotective astrocytes, which migrate towards the cortical lesion area and secrete Thbs4. Local provisional fibrinogen thus activates BMP signaling in NSPCs inducing their differentiation into astrocytes at sites of vascular permeability in the CNS. CSF cerebrospinal fluid, NSPCs neural stem/precursor cells, RGD Arg–Gly–Asp sequence (integrin binding site), SVZ subventricular zone. The Astrocyte image was adapted from http://togotv.dbcls.jp/ja/togopic.2011.17.html, under Creative Commons Attribution 4.0 International (CC BY 4.0) license. The original astrocyte image was designed by Hiromasa Ono. The Brain image was adapted from https://icon-icons.com/de/symbol/Gehirn/39333, under Creative Commons Attribution 4.0 International (CC BY 4.0) license. The original brain image was designed by Lorc, Delapouite & contributors.

contains mostly capillaries with reduced astrocyte endfeet and pericyte coverage compared to the tight BBB in other brain regions[7,37]. Astrocytes and pericytes have critical roles in the regulation of the BBB[38,39]. In pathological states, the degree of fibrinogen deposits correlated with loss of vasculature pericyte coverage[23]. However, we found that pericyte numbers were not affected after fibrinogen deposition in the SVZ niche, suggesting a dynamic, transitory alteration of the SVZ vasculature after the distant cortical injury. SVZ astrocytes and astrocyte-like B stem cells are regulators of blood flow[37,40] and pericytes have contractile functions[41]. It will be important to establish whether SVZ

astrocytes or astrocyte-like B cells through regulation of pericytes actively participate in regulating the vasculature permeability in response to CNS disease. Increased SVZ vasculature permeability with fibrinogen extravasation into the SVZ stem cell niche in animal models for cortical brain injury and stroke, as well as in human stroke patients, might be a critical event modulating stem cell behavior and contributing to brain repair.

Fibrinogen provokes astrocyte differentiation from NSPCs (this study) as well as astrocyte activation[18], but the mechanisms appear to be quite distinct. Fibrinogen is a carrier of latent TGF-beta and induces phosphorylation of Smad2 in adult astrocytes resulting in astrocyte activation and expression and secretion of the neurite outgrowth inhibitory CSPG neurocan[18]. In contrast, fibrinogen-induced newborn astrocytes did not express markers of reactive astrocytes, including Lcn, within the SVZ stem cell niche after PT (Supplementary Fig. 4b), possibly due to a lack of expression of the major activators of latent TGF-β by NSPCs, such as αvβ6 and αvβ8 integrins[42]. Indeed, NSPCs mainly express laminin-binding integrin receptors, such as α6β1 regulating NSPC behavior[43]. Future studies will show whether Fibrinogen-induced SVZ-derived Thbs4 + GFAP + newborn astrocytes that reach the cortical lesion area will become responsive to fibrinogen-bound TGF-β and whether these cells contribute to the inhibitory scar or if these newborn astrocytes will have other functions in the lesion area with a more beneficial role in brain repair.

Interestingly, fibrinogen activation of the BMP signaling pathway in NSPCs occurred independently of fibrinogen-bound BMP. Fibrinogen consists of three chains, designated Aα, Bβ, and γ and the αC domain of the fibrinogen chain Aα contains a RGD binding site that interacts with the integrin receptors α5β1, αvβ3, and αvβ8[16]. The fibrinogen αC fraction is a carrier of latent TGF-β and induces astrocyte activation and fibrinogen binding to αvβ3 integrin transactivates the EGFR in neurons[17]. NSPCs express the laminin receptor integrin α6β1[44,45] and β1-integrin signaling has been shown to regulate astrocyte differentiation from ependymal cells and NSPCs[46]. Fibrinogen colocalizes with activated integrin β1, fibrinogen-mediated NSPC differentiation into astrocytes was inhibited by blocking either integrins or by using fibrinogen isolates lacking the RGD containing αC-terminus and inhibition of lipid raft formation prevented fibrinogen-induced BMP signaling. Thus, it is tempting to speculate that fibrinogen through αC domain—β1-integrin binding enhances BMP type I receptor association in lipid rafts to activate BMP signaling and directs lineage specification. Future studies will further elucidate the molecular mechanisms of integrin-growth factor transactivation induced by fibrinogen in the stem cell niche.

In NSPCs, fibrinogen depletion reduced expression of Id3, a transcriptional regulator controlling bHLH TF activity[5], which was essential for fibrinogen-induced NSPC differentiation into astrocytes. Fibrinogen and Id2, the Id3-related protein, are upregulated in demyelinated multiple sclerosis lesions[47]. Fibrinogen inhibits the differentiation of NG2 + oligodendrocyte progenitor cells (OPC) to myelinating oligodendrocytes by promoting their differentiation to astrocyte-like cells in a BMP type I receptor-dependent manner[47]. Fibrinogen extravasation and conversion to fibrin around leaky blood vessels might allow αC domain-induced BMP signaling, leading to a rapid and robust upregulation of Id proteins, which control activity of different cell type-specific bHLH TFs. Thus, we propose that fibrinogen acts as the central player for overriding intrinsic programming of glial[48] and neuronal progenitors (this study) via upregulation of Id proteins resulting into their differentiation into astrocytes.

Astrocyte populations contributing to scar formation have been described to be heterogeneous with both beneficial and deleterious roles upon brain injury[29,49–51], including NG2 glia-

derived[48] and SVZ-derived Thbs4 + newborn astrocytes[4]. Our study showed that fibrinogen triggered SVZ-derived Thbs4 + NSPC differentiation into reactive astrocytes contributing to scar formation as well as the formation of astrocyte-like cells from NG2 + oligodendrocyte progenitor cells in a BMP type I receptor-dependent manner[47]. Thus, fibrinogen might be inhibiting neurogenesis and remyelination by BMP-dependent cell fate switch of neuronal and oligodendrocyte progenitors, respectively. Our results showed that a cortical brain insult leads to fibrinogen deposition into the SVZ and, as expected into the cortical lesion area (Fig. 1b). The cortical astrocyte scar in the lesion area after PT consists of resident and locally produced reactive astrocytes, as well as of SVZ-derived newborn astrocytes[4–6,29,35]. Fibrinogen provokes astrocyte differentiation from NSPCs via BMP receptor signaling (this study) as well as astrocyte activation by promoting the availability of active TGF-beta after vascular damage (Schachtrup et al., 2010). Astrocytes adjacent to the vasculature proliferate after CNS injury and are a major source to glial scar formation. However, depletion of fibrinogen did not affect the proliferation and formation of new (Ki67+/EdU+) local, perivascular astrocytes at the lesion site (Supplementary Fig. 8), suggesting that in the lesion area fibrinogen deposition does not affect overall astrocyte cell number, but its activation status. Injury-induced SVZ-mediated astrogenesis was recently described to be important for normal glial scar formation and BBB repair[4]. Therefore, fibrinogen cell responses on NSPCs and resident cortical astrocytes might represent a potential therapeutic target for modulating astrocyte functions and glial scar formation in CNS disease.

The discovery that a key blood-derived coagulation protein, fibrinogen, regulates the differentiation of NSPCs into astrocytes following cortical brain injury has potential implications for multiple CNS disease processes in different stem cell niches. Cortical brain injury might increase the vasculature permeability in the SVZ (this study) and the hippocampal stem cell niche[52,53]. Analysis of vasculature permeability in aging and mild cognitive impairment patients revealed that progressive BBB breakdown begins in the hippocampus and may contribute to early stages of dementia associated with AD[54]. Fibrinogen is deposited in the AD brain and its depletion protects from cognitive decline in animal models of AD[55,56]. Fibrinogen inhibited neuronal differentiation from hippocampal NSPCs and induced their differentiation into astrocytes. It will be of keen interest to determine whether fibrinogen also leaks into other stem cell niches and drives NSPC differentiation into astrocytes in other diseases as well as in aging. Therapeutic approaches of human iPSC-derived neural stem cells for transplantation show promising outcomes in improving neural repair in CNS disease[57,58]. The unfriendly environment of the lesioned brain is thought to cause an unfavorable cell differentiation with unknown function of transplanted human iPSC-derived neural stem cells. Since fibrinogen drives the SVZ NSPC differentiation into Thbs4+ reactive astrocytes (our study) with potential beneficial functions for brain repair[4], our study suggests that manipulation of fibrinogen-induced signaling pathways in human neural stem cells could control their fate and functions tailored to promote CNS repair.

## Methods

**Mice.** C57BL/6 mice (Charles River) and C57BL/6J-inbred mice deficient for $Fga^{-/-}$[59] were used. For analysis of SVZ NSPCs, Nestin-CreER[T2] mice[36] were crossed with YFP[fl] mice[60], resulting in Nestin-CreER[T2];YFP[fl] mice. C57BL/6 J-inbred mice deficient for inhibitor of DNA binding 3 (Id3[−/−])[61] were used for NSPC isolation and culture. All animal experiments were approved by the Federal Ministry for Nature, Environment and Consumers Protection of the state of Baden-Württemberg and were performed in accordance to the respective national, federal and institutional regulations.

**Cortical brain injury.** Photothrombotic ischemia (PT) was performed to induce stroke in the cortex of adult mice[21]. Briefly, 15 min post injection of the photosensitive dye Rose Bengal (Sigma–Aldrich; 10 μl/g body weight, intraperitoneally), a cold light illuminator of 150 W intensity was applied stereotaxically: (Bregma 0; mediolateral [ML], −2.4 mm according to Paxinos and Watson). The region of interest (4 mm diameter) was illuminated for 6 min, and after the light exposure was stopped, the wound was sutured. Cortical stab wound injury (SWI) was performed to induce mild traumatic injury[5,18]. To analyze fibrinogen deposition in the SVZ stem cell niche after cortical injury a 30-gauge needle was inserted stereotaxically (sagittal: anteroposterior (AP), 1.5 mm; mediolateral (ML), −0.5 mm; dorsoventral (DV), −3.0 mm; coronal: AP, 0 mm; ML, −1.2 mm; DV, −1.5 mm from bregma, according to Paxinos and Franklin) and left in place for 5 min.

**Transient focal cerebral ischemia (MCAO).** Transient focal cerebral ischemia was induced by occlusion of the middle cerebral artery (MCAO). During surgery, anesthesia was maintained with 1.3% to 1.5% isoflurane in 70% $N_2O$/30% $O_2$, and body temperature was kept constant at 37–37.5 °C using a heating pad. A silicone-coated filament was advanced through the incised left internal carotid artery until it occluded the middle cerebral artery. After 45 min MCAO, the filament was withdrawn during a second anesthesia to allow reperfusion. Animals survived for 3, 5, and 7 days before being processed for further histological analysis.

**Systemic fibrinogen depletion.** To analyze fibrinogen-induced SVZ NSPC differentiation into astrocytes and proliferation of cortical, perivascular astrocytes after PT, mice were depleted of fibrinogen with ancrod[62]. Briefly, the mice received 2,4 U ancrod or control buffer per day by mini-osmotic pumps (0.5 μL/h) implanted subcutaneously in their back. To analyze the Id3 expression in SVZ NSPCs, Nestin-CreER[T2];YFP[fl] mice were injected intraperitoneally with a daily dose of tamoxifen (TAM) (180 mg/kg/d body weight, dissolved in sunflower oil) for five consecutive days[36] and PT was performed three days after the last injection.

**EdU labeling regime.** To label the proliferating cells in the SVZ, C57BL/6 mice were intraperitoneally injected with EdU (50 mg/kg body weight, Invitrogen) immediately after PT, 6 h, 1, 2, and 3 d after PT. Mice were sacrificed 2 h after the last injection. To label newly generated juxtavascular astrocytes after photothrombotic ischemia in fibrinogen-depleted animals by ancrod, C57BL/6 mice were intraperitoneally injected with EdU (50 mg/kg body weight, Invitrogen) 5 d after PT for three consecutive times (4 h interval). Mice were sacrificed 1 day after the last injection.

**Stereotactic injection of fibrinogen.** We stereotactically injected fibrinogen[18] to investigate the direct effect of fibrinogen on NSPCs. Briefly, fibrinogen (Calbiochem) or Albumin (Sigma) was dissolved in endotoxin-free distilled water, diluted to 5 mg/ml with NaCl, and kept at 37 °C. Fibrinogen (1 μl of 5 mg/ml) or albumin (1 μl of 5 mg/ml) was slowly injected (0.2 μl/min) with a 10 μl Hamilton syringe attached to a 33-gauge needle using the same coordinates as for SWI.

**Fibrinogen preparations.** Full-length fibrinogen fraction and generation of fibrinogen lacking the intact αC region was isolated[34,63]. A thrombin-coagulable fraction lacking the intact αC region was prepared from plasma. Termed des-αC fibrinogen and evaluated by SDS-PAGE, its Bβ and γ chains were intact, as originally described[63], and by gel scanning its Aα chain core remnants ranged from 46.5–22.6 kDa (or ~66 to 32% of the intact Aα chain), thus lacking virtually all of the αC domain and variable lengths of the αC-connector. Fibrinogen was measured spectrophotometrically using the extinction coefficient (280 nm, 1 cm, 1%) of 15.5.

**Histology and immunohistochemistry.** Mice were transcardially perfused with ice-cold saline, followed by 4% PFA in phosphate buffer under ketamine and xylazine anesthesia and brain samples were removed, embedded in OCT (Tissue-Tek) and frozen on dry ice. Immunohistochemistry on coronal brain cryostat sections was performed[18]. For BrdU detection, sections were pretreated with 2 N HCl for 1 h at 37 °C. For EdU detection, we used the Click-iT™ Plus EdU Imaging kit (Fisher Scientific) according to manufacturer's protocol. For olfactory bulb morphology examination coronal olfactory bulb sections of uninjured control and $Fga^{-/-}$ brains were counterstained with hematoxylin/eosin, dehydrated in a graded ethanol series, cleared in xylene and cover-slipped with Permount (Fisher Scientific). For human sections, human brains were fixed for 2 weeks in 4% PFA and cut into 1 cm thick frontal sections. Tissues from macroscopically visible or palpable ischemic lesions as well as from the SVZ were paraffin-embedded for histopathological and immunohistochemical examinations. Samples were cut into 3 μm sections, mounted on superfrost slides and incubated at 60 °C for 30 min prior to storage at room temperature. Immunohistochemistry on brain sections was performed[18]. Corresponding tissue samples from the SVZ and the cortex of further 5 autopsy cases without significant pathological changes were collected and served as control. Primary antibodies used were rabbit anti-Aqp4 (1:300, Santa Cruz), rat anti-BrdU (1:300, Abcam), goat anti-CD13 (1:500, R&D Systems), rabbit anti-CD31 (1:50, Abcam), mouse anti-CSPG (1:200, Sigma), guinea pig anti-Doublecortin (1:1000, Millipore), rabbit anti-δ-GFAP (1:500, Millipore), rabbit anti-GFAP (1:2000, Abcam), rat anti-GFAP (1:2000,

Invitrogen), goat anti-GFP (1:2000, Abcam), rabbit anti-GFP (1:2000, Abcam), rabbit anti-Fibrinogen (1:10000, USBiological), sheep anti-Fibrinogen (1:500, USBiological), rabbit anti-Id3 (1:1000, Calbioreagent), goat anti-Lcn2 (1:1000, R&D Systems), goat anti-Nestin (1:200, Santa Cruz), goat anti-Nestin (1:500, Antibodies-Online), rabbit anti NeuN (1:1000, Abcam), rabbit anti-S100β (1:2000, Abcam), rabbit anti-Thbs4 (1:300, Genetex), sheep anti-Thbs4 (1:500, R&D Systems) and secondary antibodies used included donkey antibodies to rabbit, rat, guinea pig, mouse, sheep, and goat conjugated with Alexa Fluor 488, 594, or 405 (1:200, Jackson ImmunoResearch Laboratories). For pericyte coverage analysis, Texas red labeled lycopersicon esculentum (Tomato) Lectin (1:200, Vector Laboratories) was used concomitantly with the secondary antibodies. Sections were cover-slipped with DAPI (Southern Biotechnology).

**Culture of primary astrocytes, SVZ, and hippocampal NSPCs**. SVZ- and hippocampal-derived NSPCs were isolated and cultured[5,24]. SVZ-derived NSPCs were isolated to generate neurospheres[24]. Briefly, the dissected adult mouse SVZ tissue was dissociated with 0,25% Trypsin/HBSS and cells were cultured at a density of 50,000 cells in 25 cm$^2$ culture flasks in Neurobasal-A medium (GIBCO) containing B27 without vitamin A, Pen/Strep (1%), GlutaMax (0.5%), rhFGF2 (20 ng/ml) (all from Invitrogen) and rhEGF (20 ng/ml; Sigma–Aldrich) for generation of the non-adherent neurophere culture. For differentiation assays, SVZ NSPCs were plated on poly-D-lysine (Millipore) coated eight-well glass slides (BD Falcon) in NSPC culture medium without rhFGF2 and rhEGF. Hippocampal-derived NSPCs were isolated and cultured[64,65]. Briefly, the dentate gyrus was micro-dissected from the rest of the hippocampus under a dissection binocular microscope avoiding contamination with tissue from the molecular layer, cerebral cortex and subventricular zone. For obtaining hippocampal-derived NSPCs, the dissected dentate gyrus of the adult mouse hippocampus was dissociated with papain and plated on poly-D-lysine/laminin coated 6-well plates in DMEM/F12 + Glutamax medium containing B27 (2%), + Pen/Strep (1%), rhFGF2 (20 ng/ml) (all from Invitrogen) and rhEGF (20 ng/ml; Sigma–Aldrich). For differentiation assays, hippocampal NSPCs were plated on poly-D-lysine/laminin in culture medium with progressive removal of rhFGF2 (day 0: 10 ng/ml, day 2: 5 ng/ml, day 4: 0 ng/ml) and rhEGF (day 0: 20 ng/ml, day 2: and day 4: 0 ng/ml).

**Fibrinogen treatment of SVZ and hippocampal NSPCs**. For differentiation assays, SVZ NSPCs were plated on poly Dlysine (Millipore) coated eight well culture slides (BD Falcon) at a density of 40,000 cells per well in NSPC culture medium without rhFGF2 and rhEGF. SVZ NSPCs were treated with hirudin (0.5 Units, Sigma–Aldrich) for 15 minutes and then treated with fibrinogen (2.5 mg/ml; Calbiochem). For analysis of BMP type I receptor activation and NSPC differentiation by different fibrinogen isolates, SVZ NSPCs were treated with commercially available fibrinogen (1.0 mg/ml; Calbiochem), or with full-length fibrinogen and with fibrinogen lacking the αC domain isolated as previously described (termed DesαC, 1.0 mg/ml)[34,63]. For inhibitory assays, cells were pretreated 1 h before fibrinogen treatment with LDN193189 dihydrochloride, blocking BMP signaling by antagonizing the intracellular kinase domain of BMP type I receptors (500 nM; Sigma)[66]. For integrin-blocking peptide assays, cells were pretreated 1 h before fibrinogen stimulation with the RGDS or PDEA peptides (100 µg/ml, Sigma). SVZ NSPCs were differentiated as indicated, fixed in 4% paraformaldehyde in phosphate buffer and processed for immunocytochemistry. The hippocampal NSPCs were treated with fibrinogen (2.5 mg/ml; Calbiochem) at day 2 and day 4 after initiation of differentiation and differentiated for 7 d. For P-Smad1/5/8 analysis, hippocampal NSPCs were treated with fibrinogen at day 2 after initiation of differentiation and differentiated for 4 d. The quantification was performed as described with slight modifications[24,67]. In detail, NSPC differentiation into astrocytes and neurons was defined by morphological and antigenic properties. An arbitrary threshold was defined for the GFAP and Tuj-1 immunoreactivity as a measure for an astrocyte and neuron, respectively. For P-Smad1/5/8 analysis, nuclear immunoreactivity was measured using ImageJ.

**Immunocytochemistry**. Cells were rinsed with ice-cold PBS, fixed in 4% PFA for 30 min at 4 °C, washed three times with PBS, permeabilized for 10 min at 4 °C in PBS plus 0.1% Triton X-100 (by volume), blocked in PBS with 5% BSA for 30 min at 4 °C, and washed three times in PBS. The primary antibodies used were rabbit anti-Aldhl1l (1:1000, Abcam), rabbit anti-Aqp4 (1:300, SantaCruz); rat anti-active β1-integrin (9EG7) (1:50, BD Bioscience) rat anti-GFAP (1:2000, Invitrogen), rabbit anti-P-smad1/5/8 (1:1000, Cell Signaling), rabbit anti-Tuj-1 (1:600, Abcam), mouse anti-Tuj-1 (1:100, Millipore), and secondary antibodies used included donkey antibodies to rat, rabbit and mouse conjugated with Alexa Fluor 488, 594 or FITC (1:200, Jackson ImmunoResearch Laboratories).

**RNA isolation and quantitative PCR**. RNA was isolated from primary NSPCs and quantitative real-Time PCR was performed[68]. The following primers were used:

*Aquaporin-4*: Fwd 5'-GGAGTCAGATTACGGGCACT-3
Rev 5'-TGATCAGGTACAGGGACAATTT-3'
*Aldolase C*: Fwd 5'-AGAAGGAGTTGTCGGATATTGCT-3'
Rev 5-TTCTCCACCCCAATTTGGCTC-3'
*CyclinB1*: Fwd 5'-AAGGTGCCTGTGTGTGAACC-3'

Rev 5'-GTCAGCCCCATCATCTGCG-3'
*E2f8*: Fwd 5'-TTTTCTGAGCCACATAAAAGGGG-3'
Rev 5'-CTTCCTTGGGCTTGGTTGGT-3'
*GFAP*: Fwd 5'-ACCAGCTTACGGCCAACAG-3'
Rev 5'- CCAGCGATTCAACCTTTCTCT-3'
*GAPDH*: Fwd 5'-CAAGGCCGAGATGGGA-3'
Rev 5'-GGCCTCCCCCATTTGAT-3'
*Id3*: Fwd 5'-AAATGTTCCCTTACTGGAGGCA-3'
Rev 5'-CACAGTGCGAGAGAATGAGTTT-3'
*Smad6*: Fwd 5'- GCAACCCCTACCACTTCAGC-3'
Rev 5'-GTGGCTTGTACTGGTCAGGAG-3'.

**Gene expression profiling by microarray analysis**. RNA from primary adult NSPCs for microarray analysis was isolated with the RNeasy Micro Kit (Qiagen), according to the manufacturer's protocol. RNA samples were further processed by the Ambion WT Expression kit as described by the manufacturer (Life Technologies). Fragmentation and labeling of the amplified cDNAs was done with the GeneChip WT Terminal Labeling and Controls Kit (Affymetrix). Labeled fragments were hybridized to Affymetrix GeneChip ST 1.0 arrays for 16 h at 45 °C with 60 rpm in an Affymetrix Hybridization oven 645. After washing and staining, the arrays were scanned with the Affymetrix GeneChip Scanner 3000 7 G. CEL files were produced from the raw data with Affymetrix GeneChip Command Console Software Version 4.0. Partek Genomics Suite software (Version 6.13.0412) was employed for further analyses. CEL files were imported, including control and interrogating probes, pre-background adjustment was set to adjust for GC content and probe sequence, and RMA background correction was performed. Arrays were normalized using quantile normalization, and probe-set summarization was done using median polish. Probe values were log2 transformed. To identify differentially expressed genes between the groups, we performed a one-way ANOVA in Partek.

**Immunoblots**. For the detection of fibrinogen-induced GFAP, Aqp4 and Aldoc expression by adult NSPCs, primary cells were treated with fibrinogen (2.5 mg/ml; Calbiochem) for 2 d. For detection of P-Smad1/5/8 and Smad1, NSPCs were treated with fibrinogen (2.5 mg/ml; Calbiochem) for various timepoints. For analysis of BMP type I receptor activation and detection of P-Smad1/5/8, and Smad1 by different fibrinogen fractions, NSPCs were treated with fibrinogen (1.0 mg/ml; Calbiochem) or with fibrinogen lacking the αC domain (termed DesαC, 1.0 mg/ml)[69] for 1 h. Treatment with BMP-2 (2 ng/ml; Peprotech) and BMP-9 (10 ng/ml; Peprotech) served as positive control. For blocking experiments, adult NSPCs were treated with 2.5 mg/ml fibrinogen (Calbiochem) for 1 h and pretreated 1 h before fibrinogen treatment with the BMP type I receptor inhibitor LDN193189 dihydrochloride (500 nM; Sigma), noggin (300, 600, 900 ng/ml; R&D Technologies), or endoglin (300, 600, 900 ng/ml; R&D Technologies). The following primary antibodies were used: rat anti-GFAP (1:1000, Invitrogen), rabbit anti-aquaporin-4 (1:1000, Santa Cruz Biotechnology), rabbit anti-aldolase C (1:100, Santa Cruz Biotechnology), rabbit anti-P-Smad1/5/8, rabbit anti-Smad1, rabbit anti-GAPDH (1:1000, Cell Signaling).

**ImageJ quantification**. The Western blots were quantified using ImageJ (http://rsbweb.nih.gov/ij/) with the Gel analysis tool. Densitometry was performed with values for each band normalized to corresponding GAPDH loading controls from the same membrane.

**Microscopy and image-acquisition and analysis**. For colocalization analysis, images from sagittal and coronal brain sections were acquired with a Leica TCS SP8 laser confocal microscope with ×20 and ×40 oil immersion objectives. Images were generated from z-stack projections (0.5–1.0 µm per stack) through a distance of 15–18 µm per brain section. The colocalization of different markers was analyzed with the LAS AF analysis software by displaying the z-stacks as maximum intensity projections and using axis clipping and the rotation of the 3D-rendered images. For immunoreactivity (IR) analysis, 10–18 µm projection z-stacks were saved as TIFF. With ImageJ (NIH), the images were converted into black and white 8-bit images and thresholded. Total IR was calculated as percentage area density defined as the number of pixels (positively stained areas) divided by the total number of pixels (sum of positively and negatively stained area) in the imaged field. For colocalization and immunoreactivity analysis, the images were performed on an area of 290 × 290 in the cortex and the hippocampus, 125 × 125 in the dorsal horn and 200 × 10 in the SVZ and around the 3rd ventricle. In each animal and patient, at least three brain sections were randomly selected at the level of the cortical lesion, analyzed and averaged. For colocalization analyses of YFP and NeuN, immunoreactivity images of an area of 387.5 × 387.5 µm in the glomerular layer of the OB were acquired in green and red channels simultaneously. For measurements of OB size images were acquired with a Leica MZ 10 F stereomicroscope and a Leica MC 120HD camera. Evaluation of OB length was performed with ImageJ (NIH). For cell culture assays, images used for quantifications were acquired with an Axioplan 2 Imaging epifluorescence microscope with ×20 objective and analyzed with the AxioVision image analysis software (Carl Zeiss). The images were saved as TIFF and quantified with the Cell counter plugin of ImageJ. The immunoreactivity measurements were performed by using ImageJ, as described above. To quantify nuclear P-Smad1/5/8 immunoreactivity in the

hippocampal NSPCs, every DAPI + nuclei were selected and quantified. At least 80 nuclei per image were quantified and the P-Smad1/5/8 immunoreactivity values were averaged. The representative images were acquired with a Leica TCS SP8 laser confocal microscope with ×20 and ×63 oil immersion objectives. Images were generated from z-stack projections (0.35–1.0 μm per stack) through a distance of 1–12 μm.

**Statistics**. Data are shown as means ± SEM. Differences between groups were examined by one-way ANOVA for multiple comparisons, followed by Bonferroni's correction for comparison of means or by unpaired two-tailed or one-tailed Student's t-test to analyze significance between two experimental groups using GraphPad Prism (GraphPad Software).

**Reporting summary**. Further information on research design is available in the Nature Research Reporting Summary linked to this article.

## Data availability
Microarray data reported in this paper are available at ArrayExpress data base with accession number E-MTAB-3013. Source data are provided as a Source Data file. All other data are available from the corresponding author upon reasonable request.

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

## Acknowledgements

We thank Meike Ast-Dumbach for outstanding technical assistance, A. Schober for graphics, Sagar and Bernhard Kramer for excellent imaging support, Matthias Kirsch and Jan Pruszak for reagents and experimental support, Jay L. Degen for providing the *fgα*$^{-/-}$ mice, and the University of Freiburg Live Imaging Center (LIC) for microscopy support. This study was supported by the International Graduate Academy Fellowship (State Baden-Württemberg) to C.B., by a Deutscher Akademischer Austauschdienst fellowship to K.M., by a Fazit Foundation Graduate fellowship to S.S., by the Department of Defense (DoD) MS160082 and NIH/NINDS R35 NS097976 Grants to K.A, the European Commission FP7 Grant PIRG08-GA-2010-276989 and the German Research Foundation Grant SCHA 1442/5-1, and 1442/6-1 to C.S.

## Author contributions

L.P. performed the majority of the experiments. S.S.D. and C.B. performed surgeries. S.N., S.M., S.C.M., S.S., C.B. and K.T. contributed to histology, immunocytochemistry, and biochemical experiments. D.P. performed microarray analysis. S.D., F.F.K., D.K.G., V.T. and K.A. provided crucial reagents and tissue samples, D.K.G., V.T. and K.A. contributed to experimental design, data analysis and interpretation. C.S. designed the study, analyzed data, coordinated the experimental work, and wrote the manuscript with contribution from all authors.

## Competing interests

The authors declare no competing interests.
