## [Peer Review File · Nature Communications]

Editorial Note: Parts of this peer review file have been redacted as indicated to remove third-party material where no permission to publish could be obtained.

Reviewers' comments:

Reviewer #1 (Remarks to the Author):

This interesting paper by Pous, et al describes the role of Fibrinogen in promoting astrogenesis in SVZ-NSC populations after injury. Previous work from other labs has shown that cortical injury can induce protective astrogenesis from SVZ populations. Focusing on fibrinogen, a compound in the blood that is released into the brain parenchyma after injury and can modulate a host of cellular responses, they show that it is upregulated in the SVZ after injury. This upregulation of Fibrinogen is correlated with the induction of protective astrogenesis in vivo and they show in vitro that it is sufficient to promote astrocyte formation from NSC populations. Next, they manipulate Fibrinogen expression and show changes in SVZ marker expression that infers a loss of stem cells capacity and suggests a bias towards astrocyte production. Mechanistically, they identify increases in BMP signaling (and Id3) as a potential fibrinogen-induced mechanism driving astrocyte generation. This is an interesting paper that tackles a critical area of injury response: the generation of protective astrocytes from the SVZ nice. Despite the critical topic covered in this paper, there are obvious flaws in the experiments and the organization of the paper that need correction before this paper is suitable for publication. Below are some comments meant to improve this manuscript:

- 1) The biglycan and CNTF aspects of the story are tangential and make for an unfocused conclusion to the paper. The crux of this story is Fibrinogen-BMP signaling in the SVZ and how this axis promotes protective astrogenesis after injury. These last parts of the paper are not necessary
- 2) My main issue experimentally is that they do not show whether the production of reactive astrocytes at the injury site is impaired in the Fib-KO or that these impairments are the result of defective production of astrocytes from the SVZ. They clearly show a shift in the GFAP/s100b marker balance early after injury, but they never show whether responses at the injury site are altered in the KO. This is critical, because as it stands we have no idea whether Fibrinogen in the SVZ is actually important for injury responses. There is an inference based on other work, but this is not rigorously proven.
- 3) To follow this point, in Figure 1 they show Fibrinogen deposition all over the brain after PT injury. It's likely that fibrinogen is promoting local astrocyte proliferation and thus reactive astrocyte responses at the local injury site. Might be a good idea to compare the local versus SVZ effects of fibrinogen. Also, given that they have pharmacological methods for manipulating fibrinogen, they should consider inhibiting fibrinogen specifically in the SVZ after injury and assessing reactive astrocyte responses at later timepoints. In this way, they can decipher the role fibrinogen specifically on SVZ populations after injury.
- 4) Towards this, it would be good to show that the number of DCX-expressing cells that migrate to the injury site is altered in the Fib-KO. Also, GFAP/S100b marker bias is an antiquated method for sub-typing SVZ cells. Thbs4 has been shown to be specifically induced in SVZ cells after injury. I would recommend assessing the Thbs4-Notch axis and how it relates to Fibrinogen
- 5) Is Id3 expression upregulated in the SVZ after PT? Along these same lines, they should consider pharmacologically inhibiting BMP signaling in the SVZ after injury and assess reactive astrocyte responses at the local site of injury.

Reviewer #2 (Remarks to the Author):

This manuscript by Pous et al. reports that the blood derived factor fibrinogen increases the propensity of subventricular zone stem cells to differentiate into astrocytes. They demonstrate GFAP reactivity and fibrinogen deposition in the SVZ after three injury models, as well as in human stroke. They show that exogenous fibrinogen in vitro increased GFAP, and fibrinogen deficiency in vivo (genetic and pharmacological) decreased GFAP positive cells in the SVZ. The authors then perform a series of in vitro studies, including bulk transcriptomics (Fig 3-5) arguing that BMP signaling is a possible target of

fibrinogen signaling, that it is downstream of the TF Id3, and that the receptor in this signaling cascade could be B1 integrins.

Understanding the external factors regulating neural stem cell fate choice is an important topic. Although infiltration of fibrinogen into the brain after injury is well established, its impact on neural progenitors has not been studied and has important potential implications. The downstream pathways identified by the authors make sense and are good candidates. However, the impact of this study is limited in that many of the conclusions are drawn from in vitro work, and the in vivo experiments are too limited to draw firm conclusions about what all of this means for the brain. Specifically, because GFAP is a marker of some murine astrocytes, but mostly of reactive astrocytes, there is no reason to think that an increase in GFAP in the SVZ reflects a change in stem cell fate. The more likely explanation is that it reflects reactivity, as one would expect after an injury. In addition, since gliogenesis often occurs at the expense of neurogenesis, it is striking that a study purporting to examine SVZ stem cell fate does not quantify olfactory bulb neurogenesis in vivo with a BrdU pulse labeling assay. In summary, while this is a potentially important work and the data flow logically, the approaches used limit the conclusions that can be drawn.

Some specific comments:

Fig 1: In Fig 1, they demonstrate GFAP reactivity and fibrinogen deposition in the SVZ after photothrombotic lesion, stab wound, and MCAO, as well as in human stroke. The authors argue based on localization that fibrinogen enters from blood vessels rather than the CSF, which seems reasonable but is difficult to prove. This language could be softened.

Fig 4- The authors suggest that fibrinogen colocalizes and interacts with B1 integrin, an NSC marker, using a fractionated serum derived fibrinogen. Recombinant proteins lacking specific domains would certainly have been more elegant, as it is difficult to know what else might be different about this plasma fraction. Serum very potently drives astrogenesis and inhibits neurogenesis in vitro.

Fig 5 shows upregulation of biglycan protein after fibrinogen, although the significance of this is unclear, as it is just another astrocyte marker, thus not necessarily surprising.

Fig 5 also shows a pro astrogenic effects of fibrinogen on hIPSCs- quantification in 5c does not show the most relevant statistical result, which is control vs fibrinogen in the presence of CNTF

Reviewer #3 (Remarks to the Author):

In this manuscript the authors continue their studies on the role of fibrinogen in the generation and activation of astrocytes in the CNS. Specifically, the current manuscript investigates the effects of fibrinogen deposition in the subventricular zone following induction of a stroke or stab wound model. The primary model is a photothrombotic ischemia that induces significant unilateral damage to the CNS. The authors demonstrate that rapidly following stroke there is an increase in fibrinogen in the SVZ and this is correlated with an up-regulation of markers of mature astrocytes (GFAP etc.). The authors propose this is a consequence of induction of astrocyte fate in neural stem cells in the SVZ. To confirm their hypothesis the authors extend the study to an in vitro culture model and define the signaling pathways as being mediated through activation of the BMP pathway. Similar treatment of human iPS derived NSC resulted in a similar outcome in terms of the elevation of the number of mature astrocytes, while microarray analysis suggested the astrocytes that were generated by this paradigm had neuroprotective characteristics.

Understanding the sequelae of CNS injury of neural cell fate is an important and timely issue. This manuscript clearly demonstrates that following a CNS insult there is a dramatic increase in the number of GFAP+ cells adjacent to germinal zones and at least part of that increase depends on the

presence of fibrinogen. The manuscript in its current form does not, however provide compelling evidence that the increase in GFAP+ cells reflect fate changes in NSCs. Furthermore, the biological significance of the observations is missing. While the authors might suggest that induction of astrocytes enhances or inhibits CNS recovery this is not shown and it is equally possible that there is no change in pathology. In this case, while the biology may be interesting the impact of this manuscript is limited.

Some additional issues that need to be addressed.

1. The issue of induced cell fate. The paper is very weak on descriptions of NSC. The quoted manuscript (Lugert et al) used a transgenic animal and FACs sorting – the current manuscript uses wild type animals.
2. The characterization of the NSCs is missing. This is critical since the data may also be explained by the maturation of existing astrocyte/oligodendrocyte precursors (with no true fate switch). Indeed, this group has demonstrated such an effect in their 2017 neuron paper.
3. The data in Fig 2 is difficult to interpret without data showing total cell number. Likewise, there is no attempt to examine cell proliferation or cell death either in vivo (Fig 2F) or in vitro.
4. The human iPS data seems very superficial and adds little to the overall story.

Point-by-Point Reply

Our manuscript entitled “**Fibrinogen induces neural stem cell differentiation into astrocytes in the subventricular zone stem cell niche via BMP receptor signaling**” has been reviewed by three experts in the field. The authors would like to kindly thank the Reviewers for their constructive comments and helpful suggestions. Prompted by the constructive and insightful comments of reviewers 1, 2, and 3, we performed additional experiments to further characterize the role of fibrinogen in NSPC differentiation into astrocytes.

In particular, we experimentally addressed the two major points of the Reviewers, namely:

- 1) How fibrinogen alters the lesion pathology after photothrombotic ischemia?
- 2) What is the mechanism of fibrinogen altering the cell fate of NSPCs and what is the biological significance of fibrinogen-induced NSPC differentiation into astrocytes after photothrombotic ischemia?

Briefly, our revised manuscript provides the following insights to these two important questions:

1) To analyze the role of fibrinogen in the lesion pathology, we acutely depleted fibrinogen from the circulation by anicrod and analyzed cortical scar formation in the photothrombotic ischemia (PT) mouse model of stroke. Anicrod-treated animals showed a 93% reduction of fibrinogen levels in the cortical lesion area compared to vehicle-treated control animals (**data not shown**). As shown previously, astrocyte activation contributing to scar formation in the nervous system begins within hours after traumatic injury and is characterized primarily by reactive astrocytes upregulating GFAP expression and depositing proteins of the chondroitin sulfate proteoglycan family (CSPGs) that are inhibitory for regeneration (Schachtrup et al., 2010). In the revised manuscript, we provide striking new data that fibrinogen depletion by anicrod drastically reduced the GFAP and CSPG expression level in the penumbra by 75% and 72%, respectively, confirming that fibrinogen was necessary for astrocyte scar formation (**Fig. 5a-b in the revised manuscript**). These new data demonstrate that fibrinogen alters cortical lesion pathology.

2) We further experimentally addressed how fibrinogen alters the cell fate of SVZ NSPCs and how fibrinogen-induced SVZ-derived newborn astrocytes contribute to the lesion area and found that:

- Fibrinogen induced the proliferation of SVZ Thbs4⁺ NSPCs, resulting in an increased total number of the Thbs4⁺ cells on day 3 after PT (**Fig. 2g in the revised manuscript, Supplementary Fig. 4d**). These Thbs4⁺ cells further differentiated into S100 β +GFAP⁺ mature astrocytes within the SVZ (**Fig. 2h in the revised manuscript**). Fibrinogen depletion by anicrod or by using fibrinogen deficient mice (*Fga*^{-/-} mouse line) drastically reduced the number of S100 β +GFAP⁺ mature astrocytes within the SVZ (**Fig. 2i and Supplementary Fig. 4e in the revised manuscript**).
- Fibrinogen depletion drastically reduced the number of Thbs4+GFAP⁺ astrocytes that had migrated into the penumbra by 74% compared to control mice 6 days after PT (**Fig. 5c in the revised manuscript**).
- To differentiate between local cells and SVZ-derived newborn astrocytes, we added cell tracking experiments using tamoxifen inducible *Nestin-CreER^{T2};YFP^{fl}* reporter transgenic mice (Lagace et al., 2007). Remarkably, immunolabeling of SVZ-derived YFP⁺ cells in combination with Thbs4 and GFAP in the lesion area revealed a 64.5% reduction of SVZ-derived astrocytes in anicrod treated animals compared to control animals 10 days after PT, while the overall number of SVZ-derived YFP⁺Thbs4⁺ cells at the lesion scar was not changed (**Fig. 5d in the revised manuscript**).
We performed additional experiments using the 5-bromo-2'-deoxyuridine (BrdU) labeling regime for cell tracking in combination with elimination of fibrinogen using the *Fga*^{-/-} mouse line. Those showed, in accordance to the anicrod data, a 50% reduction in the number of SVZ-derived BrdU⁺GFAP⁺ reactive astrocytes in the lesion area compared to control mice in the stab wound injury model (**Supplementary Fig. 9a-c**).
- Fibrinogen was sufficient to induce SVZ adult NSPC differentiation into SVZ-derived BrdU⁺GFAP⁺ astrocytes by stereotactically injecting fibrinogen into the cortex in close proximity to the SVZ (**Supplementary Fig. 9d-e**).

Overall, these results reveal that in the SVZ fibrinogen has physiological relevance as fibrinogen deposition in the SVZ massively affects the distal cortical scar formation by affecting SVZ-derived NSPCs. In particular, we showed that fibrinogen drives newborn SVZ-derived Thbs4+ astrocyte differentiation in the lesion area.

Overall, we have **revised Figures 3c-f, and 4c-d**, added to the manuscript a **new Figure 5** and three new Supplementary Figures (**Supplementary Figures 8-10**), added six new panels (**Figure 2g-h, Figure 4f, Supplementary Figure 3b, f-g, Supplementary Figure 4b-d**), additional quantifications (**Figure 4c**) and improved the labeling, as requested by the Reviewers.

Our new experiments have bolstered our original findings on the novel mechanism that fibrinogen drives NSPC differentiation into astrocytes within the SVZ and the contribution of SVZ-derived newborn astrocytes to the lesion scar formation. Our findings are a significant advance in the field that will launch new areas of research and foster new questions regarding blood-derived factors on neural stem cells, and the functions of these newborn astrocytes. A detailed point-by-point reply follows below and all changes in the revised manuscript are underlined.

Point-by-Point

Referee #1: This by Pous, et al describes the role of Fibrinogen in promoting astrogenesis in SVZ-NSC populations after injury. Previous work from other labs has shown that cortical injury can induce protective astrogenesis from SVZ populations. Focusing on fibrinogen, a compound in the blood that is released into the brain parenchyma after injury and can modulate a host of cellular responses, they show that it is upregulated in the SVZ after injury. This upregulation of Fibrinogen is correlated with the induction of protective astrogenesis in vivo and they show in vitro that it is sufficient to promote astrocyte formation from NSC populations. Next, they manipulate Fibrinogen expression and show changes in SVZ marker expression that infers a loss of stem cells capacity and suggests a bias towards astrocyte production. Mechanistically, they identify increases in BMP signaling (and Id3) as a potential fibrinogen-induced mechanism driving astrocyte generation.

This is an interesting paper that tackles a critical area of injury response: the generation of protective astrocytes from the SVZ mice. Despite the critical topic covered in this paper, there are obvious flaws in the experiments and the organization of the paper that need correction before this paper is suitable for publication. Below are some comments meant to improve this manuscript:

1. The biglycan and CNTF aspects of the story are tangential and make for an unfocused conclusion to the paper. The crux of this story is Fibrinogen-BMP signaling in the SVZ and how this axis promotes protective astrogenesis after injury. These last parts of the paper are not necessary.

In agreement with the Reviewer and to better focus the manuscript on the role of fibrinogen – BMP signaling in the SVZ and its contribution to brain repair we removed the human iPSC-derived progenitor cell aspects (**Fig. 5 of the original manuscript**), as these experiments are not necessary to understand the role of fibrinogen on SVZ NSPC differentiation into astrocytes.

We moved the biglycan data to the **Supplementary Fig. 6 of the revised manuscript** to emphasize that fibrinogen-induced newborn astrocytes are not merely activated astrocytes, but rather a unique cell type secreting neuroprotective factors. Our new experiments showed that fibrinogen promotes Thbs4+ NSPC proliferation and their differentiation into mature astrocytes. These fibrinogen-driven SVZ-derived Thbs4+ astrocytes in the lesion area might have a beneficial role in brain repair via the Thbs4-Notch axis, as previously shown (Benner et al., 2013) and via fibrinogen-induced biglycan expression and secretion (**our study, Supplementary Fig. 6b**), a factor that has been shown to promote neuronal survival (Koops et al., 1996).

2. My main issue experimentally is that they do not show whether the production of reactive astrocytes at the injury site is impaired in the Fib-KO or that these impairments are the result of defective production of astrocytes from the SVZ.

They clearly show a shift in the GFAP/S100b marker balance early after injury, but they never show whether responses at the injury site are altered in the KO. This is critical, because as it stands we have no idea whether Fibrinogen in the SVZ is actually important for injury responses. There is an inference based on other work, but this is not rigorously proven.

We thank the author for his insightful comments and suggestions. We performed several new experiments to address, whether fibrinogen-induced SVZ astrocytes contribute to the lesion responses.

We provide striking new data that fibrinogen depletion by anicrod drastically reduced the GFAP and CSPG expression level in the penumbra by 75% and 72%, respectively (**Fig. 5a-b in the revised manuscript**), revealing that fibrinogen alters lesion pathology.

Next, we investigated in detail the cell responses of the NSPC population in the SVZ after PT and how the newly generated SVZ astrocytes contributed to the lesion area. We found that fibrinogen induces the proliferation of the SVZ Thbs4+ NSPCs and their total number on day 3 after PT (**Fig. 2g in the revised manuscript, Supplementary Fig. 4d**). These Thbs4+ cells then further differentiate into S100 β +GFAP+ mature astrocytes within the SVZ (**Fig. 2h in the revised manuscript**). Importantly, fibrinogen depletion drastically reduced the number of S100 β +GFAP+ mature astrocytes within the SVZ (**Fig. 2i and Supplementary Fig. 4e in the revised manuscript**). Interestingly, we neither identified mature astrocyte proliferation in the first days after PT (**Supplementary Fig. 4c, bottom in the revised manuscript**) nor reactive astrocytes expressing activation marker, including Lcn2, Serpina3n or CSPGs within the SVZ (**Supplementary Fig. 4b in the revised manuscript and data not shown**).

Next, we investigated whether fibrinogen-induced SVZ-generated Thbs4+ astrocytes migrated to the cortical lesion area and contributed to the glia scar after PT. Strikingly, our new data showed that fibrinogen depletion drastically reduced the number of SVZ-derived Thbs4+GFAP+ astrocytes in the penumbra by 74% compared to control mice 6 days after PT (**Fig. 5c in the revised manuscript**). Furthermore, we investigated the role of fibrinogen on the cell fate of SVZ-derived NSPCs by cell tracking using tamoxifen inducible *Nestin-CreER^{T2};YFP^{fl}* reporter transgenic mice or by using the 5-bromo-2'-deoxyuridine (BrdU) labeling regime. Remarkably, immunolabeling of SVZ-derived YFP+ cells in combination with Thbs4 and GFAP in the lesion area revealed a 64.5% reduction of SVZ-derived astrocytes in anicrod treated animals compared to control animals 10 days after PT (**Fig. 5d in the revised manuscript**), while the overall number of SVZ-derived YFP+Thbs4+ cells at the lesion scar was not changed, further substantiating our findings that fibrinogen-induced SVZ-generated Thbs4+ astrocytes migrated to the cortical lesion area and contributed to the glia scar after PT.

We performed additional experiments by elimination of fibrinogen using the *Fga^{-/-}* mouse line and cell tracking using BrdU labeling. These experiments showed, in accordance to the anicrod data, a 50% reduction in the number of SVZ-derived BrdU+GFAP+ reactive astrocytes in the lesion area compared to control mice in the stab wound injury model (**Supplementary Fig. 9a-c**). Finally, we showed that fibrinogen was sufficient to induce SVZ adult NSPC differentiation of BrdU+ cells into BrdU+GFAP+ astrocytes by stereotactically injecting fibrinogen into the cortex in close proximity to the SVZ (**Supplementary Fig. 9d-e**).

Overall, these new data further bolster our finding that in the SVZ stem cell niche fibrinogen induces Thbs4+ NSPC proliferation and differentiation and that these newly formed SVZ-derived astrocytes significantly contribute to the cortical lesion area. Deposition of fibrinogen in the SVZ after cortical injury has physiological relevance as this massively affects the distal cortical scar formation (change in pathology) by affecting SVZ NSPCs.

3) To follow this point, in Figure 1 they show Fibrinogen deposition all over the brain after PT injury. It's likely that fibrinogen is promoting local astrocyte proliferation and thus reactive astrocyte responses at the local injury site. Might be a good idea to compare the local versus SVZ effects of fibrinogen. Also, given that they have pharmacological methods for manipulating fibrinogen, they should consider inhibiting fibrinogen specifically in the SVZ after injury and assessing reactive astrocyte responses at later timepoints. In this way, they can decipher the role fibrinogen specifically on SVZ populations after injury.

Our new experiments showed that fibrinogen depletion by anicrod drastically changed the glial scar formation (please see response above, **Fig. 5a-b in the revised manuscript**) on day 6 after PT. Astrocytes adjacent to the vasculature proliferate after CNS injury and contribute to glial scar formation (Bardehle et al., 2013). We analyzed whether local astrocyte proliferation or fibrinogen-induced newly generated SVZ astrocytes contributed to the lesion area and our new data showed that depletion of fibrinogen did not affect local, perivascular astrocyte proliferation at the lesion site (**Supplementary Fig. 8**), revealing that fibrinogen does not affect perivascular astrocyte proliferation.

Next, we performed new experiments to unravel the contribution of fibrinogen-induced newly generated SVZ astrocytes to the lesion area. Fibrinogen in the SVZ stem cell niche induces Thbs4+ NSPC proliferation and differentiation (**please see response above**). Our new data showed that Thbs4+ NSPCs proliferate and further differentiate into S100 β +GFAP+ mature astrocytes within the SVZ (**Fig. 2h in the revised manuscript**). Strikingly, our new experiments showed that fibrinogen depletion drastically reduced the number of SVZ-derived Thbs4+GFAP+ astrocytes in the penumbra by 74% compared to control mice 6 days after PT (**Fig. 5c in the revised manuscript**) and by using tamoxifen inducible *Nestin-CreER^{T2};YFP^{fl}* reporter transgenic mice, our new data revealed a 64.5% reduction of SVZ-derived YFP+Thbs4+GFAP+ astrocytes in anicrod treated animals compared to control animals 10 days after PT (**Fig. 5d in the revised manuscript**),

Importantly, genetic (day 3 after PT) or pharmacologic depletion of fibrinogen (day 6 after PT) significantly reduced the number of newborn S100 β +GFAP+ astrocytes in the SVZ (**Fig. 2i, Supplementary Fig. 4e in the revised manuscript**), suggesting that in the SVZ fibrinogen triggers NSPCs to differentiate into astrocytes. Subsequently, they migrate to the lesion area and these cells then contribute to the GFAP+ and S100 β + cortical astrocyte population in the lesion area (**Fig. 5 in the revised manuscript**). It is possible that in the cortical lesion area fibrinogen might then further trigger the maturation/activation of these SVZ-derived NSPCs.

4) Towards this, it would be good to show that the number of DCX-expressing cells that migrate to the injury site is altered in the *Fib-KO*.

We thank the Reviewer for this constructive comment. We performed new experiments to examine SVZ-derived DCX+ neuroblasts at the lesion site by using the *Nestin^{CreERT2};Rosa26^{YFP}* mouse line. We observed only a small population of YFP+DCX+ cells reaching the lesion area (~1% of YFP+ cells, data not shown), which is in accordance with the literature that the SVZ response towards the cortical lesion area is mainly astrogenic (Faiz et al., 2015; Bohrer et al., 2015; Benner et al., 2013; Arvidsson et al., 2002). We then depleted fibrinogen by anicrod and did not observe an increase in the number of SVZ-derived DCX+ neuroblasts reaching the lesion area. We performed additional experiments and investigated, whether fibrinogen affects SVZ-derived NSPCs reaching the olfactory bulbs after PT by cell tracking using tamoxifen inducible *Nestin-CreER^{T2};YFP^{fl}* reporter transgenic (**Supplementary Fig. 10 in the revised manuscript**). Our new experiments analyzing SVZ-derived DCX+ neuroblasts in the ipsilateral olfactory bulb revealed a 2.6-fold increase in YFP+ cells and a 1.6-fold increase in YFP+NeuN+ cells in fibrinogen-depleted animals compared to control animals 10 days after PT (**Supplementary Fig. 10 in the revised manuscript**).

Our new data suggest that in the SVZ niche area fibrinogen indeed drives NSPC differentiation into astrocytes at the expense of olfactory bulb neurogenesis after PT. We suggest that, after a cortical lesion, newborn DCX+ neuroblasts on their way to the lesion area are especially sensitive to the changed environment. These cells have to overcome, in addition to fibrinogen, many obstacles in the injury induced

unfriendly environment to survive, migrate and integrate on their way from the SVZ niche area to the cortical lesion area, which might be the reason for the low number of SVZ-derived DCX+ cells reaching the lesion area despite fibrinogen depletion. We added new sentences to the result section on **pages 13 in the revised manuscript**.

Also, GFAP/S100 β marker bias is an antiquated method for sub-typing SVZ cells.

The S100 β marker has been shown to be absent from GFAP-expressing cells of the SVZ and its onset of expression characterizes a terminal maturation stage of cortical astrocytic cells (Raponi *et al*, 2007). Nevertheless, we observed a significantly increased number of S100 β +GFAP+ cells in the SVZ after ischemic stroke. More recently, Thbs4 has been described as a marker of a SVZ NSPC subpopulation giving rise to neuroprotective astrocytes after brain injury (Benner *et al.*, 2013). By using this marker in combination with GFAP and S100 β , we observed that the Thbs4+ population proliferates and generates mature astrocytes (Thbs4+GFAP+S100 β +) in the SVZ after cortical injury (**Fig. 2g-h in the revised manuscript**).

Furthermore, we used additional markers for reactive astrocytes in the SVZ (Zamanian *et al.*, 2012), including Lcn2, Serpina3n or CSPGs and surprisingly, PT and fibrinogen deposition within the SVZ did not induce the abundance of cells expressing these markers (**Supplementary Fig. 4b in the revised manuscript and data not shown**).

Thbs4 has been shown to be specifically induced in SVZ cells after injury. I would recommend assessing the Thbs4-Notch axis and how it relates to Fibrinogen

Our new experiments revealed that the number of Thbs4+ cells increased after PT and that these cells further differentiated into Thbs4+GFAP+S100 β + astrocytes within the SVZ (**Fig. 2. g-h in the revised manuscript**). Importantly, fibrinogen depletion by anicrod significantly reduced Thbs4+GFAP+ SVZ-derived astrocytes at the lesion site (**Fig. 5c in the revised manuscript**), revealing that fibrinogen might drive the maturation and activation of these SVZ-derived newborn astrocytes. Finally, by using the transgenic mouse line *NestinCreERT2:Rosa26YFP* and pharmacologic depletion of fibrinogen, we revealed that indeed SVZ-derived Thbs4+ cells failed to differentiate into mature Thbs4+GFAP+ astrocytes contributing to the lesion area (**Fig. 5d in the revised manuscript**). In addition, we added a sentence to the discussion (**page 17 in the revised manuscript**), suggesting that the fibrinogen driven SVZ-derived Thbs4+ astrocytes in the lesion area might have a beneficial role in brain repair via the Thbs4-Notch axis, as previously shown (Benner *et al.*, 2013).

5) Is Id3 expression upregulated in the SVZ after PT? Along these same lines, they should consider pharmacologically inhibiting BMP signaling in the SVZ after injury and assess reactive astrocyte responses at the local site of injury.

Our original manuscript demonstrated that fibrinogen depletion reduced the YFP+Id3+ cell numbers compared to control treated animals after PT (**Fig. 3f in the original manuscript**). To further assess the role of the BMP-Id3 axis, we performed new experiments showing that Id3 was rapidly upregulated in YFP+ stem cells in the SVZ 1 day after PT compared to the uninjured control, while pharmacologic depletion of fibrinogen reduced Id3 expression in YFP+ stem cells in the SVZ compared to control animals after PT (**Fig. 3f in the revised manuscript**). This suggests that fibrinogen deposition in the SVZ stem cell environment after cortical brain injury rapidly activated the BMP receptor signaling pathway upregulating Id3, which in turn promotes NSPC differentiation into astrocytes. We added sentences to the result section on **page 11** of the **revised manuscript**. Our lab previously showed that the transcriptional regulator Id3 is one of the downstream effectors of the BMP signaling pathway, which regulates BMP induced NSPC differentiation into astrocytes (Bohrer *et al.*, 2015). We had demonstrated previously that genetic depletion of *Id3* in BrdU-injected mice led to a significant decrease in the number of BrdU+GFAP+ astrocytes in the lesion area 10 days after SWI (**Fig. R1, below**) (Bohrer *et al.*, 2015).

In accordance, primary *Id3*^{-/-} NSPCs are resistant to fibrinogen-induced NSPC differentiation into astrocytes (**Fig. 3e in the revised manuscript**). In conclusion, these results highlight that immediate fibrinogen deposition in the SVZ stem cell niche and its activation of the BMP receptor signaling pathway and upregulation of the transcriptional regulator *Id3* might regulate the NSPC differentiation into astrocytes and their contribution to the lesion scar. We added our thoughts to the discussion on **pages 16-17 in the revised manuscript stating that** “fibrinogen acts as the central player for overriding intrinsic programming of glial (Hackett et al., 2018) and neuronal progenitors (this study) via upregulation of *Id* proteins resulting into their differentiation into astrocytes.”

Reviewer #2: This manuscript by Pous et al. reports that the blood derived factor fibrinogen increases the propensity of subventricular zone stem cells to differentiate into astrocytes. They demonstrate GFAP reactivity and fibrinogen deposition in the SVZ after three injury models, as well as in human stroke. They show that exogenous fibrinogen in vitro increased GFAP, and fibrinogen deficiency in vivo (genetic and pharmacological) decreased GFAP positive cells in the SVZ. The authors then perform a series of in vitro studies, including bulk transcriptomics (Fig 3-5) arguing that BMP signaling is a possible target of fibrinogen signaling, that it is downstream of the TF Id3, and that the receptor in this signaling cascade could be B1 integrins.

Understanding the external factors regulating neural stem cell fate choice is an important topic. Although infiltration of fibrinogen into the brain after injury is well established, its impact on neural progenitors has not been studied and has important potential implications. The downstream pathways identified by the authors make sense and are good candidates. However, the impact of this study is limited in that many of the conclusions are drawn from in vitro work, and the in vivo experiments are too limited to draw firm conclusions about what all of this means for the brain.

Specifically, because GFAP is a marker of some murine astrocytes, but mostly of reactive astrocytes, there is no reason to think that an increase in GFAP in the SVZ reflects a change in stem cell fate. The more likely explanation is that it reflects reactivity, as one would expect after an injury.

We thank the author for his insightful comments. We performed additional experiments to further strengthen that fibrinogen indeed alters the cell fate choice of SVZ NSPCs:

Originally, we found a significant increase in GFAP+S100 β ⁺ cells in the SVZ 3 and 7 days after PT (**Fig. 2f of the original manuscript**). To identify whether SVZ astrocytes or NSPCs contributed to this pool of cells, our new experiments analyzed in detail the cell responses of the NSPC population in the SVZ at different timepoints after PT and found that fibrinogen induces the proliferation of the SVZ Thbs4⁺ NSPCs and their total number on day 3 after PT (**Fig. 2g in the revised manuscript, Supplementary Fig. 4d**).

This Thbs4⁺ cells then start to further differentiate into S100 β +GFAP⁺ mature astrocytes within the SVZ (**Fig. 2h in the revised manuscript**). Importantly, fibrinogen depletion by anicrod and using the fibrinogen deficient mouse line (*Fga*^{-/-} mouse line) drastically reduced the number of S100 β +GFAP⁺ mature astrocytes within the SVZ, respectively (**Fig. 2i and Supplementary Fig. 4e in the revised manuscript**).

The already mature resident SVZ GFAP+S100 β ⁺ astrocytes did not proliferate during the first days after PT, as shown by absent EdU labeling, suggesting that these cells are not the source for the increased number of astrocytes in the SVZ after PT (**Supplementary Fig. 4c in the revised manuscript**). Interestingly, we could not identify reactive astrocytes expressing activation marker, including Lcn2, Serpina3n or CSPGs within the SVZ (**Supplementary Fig. 4b in the revised manuscript and data not shown**), suggesting that Fibrinogen-induced NSPC differentiation into astrocytes within the SVZ is responsible for the increased number of GFAP+S100 β ⁺ cells after PT.

Furthermore, our new experiments showed that the fibrinogen-induced SVZ-generated Thbs4⁺ astrocytes migrated to the cortical lesion area and contributed to the glia scar after PT. We provide striking new data showing that fibrinogen depletion by anicrod drastically reduced the GFAP and CSPG expression level in the penumbra by 75% and 72%, respectively (**Fig. 5a-b in the revised manuscript**), revealing that fibrinogen alters lesion pathology. Astrocytes adjacent to the vasculature proliferate after CNS injury and contribute to glial scar formation (Bardehle et al., 2013). However, depletion of fibrinogen did not affect local, perivascular astrocyte proliferation at the lesion site, as shown by Ki-67 quantification (**Supplementary Fig. 8**). Strikingly, fibrinogen depletion drastically reduced the number of SVZ-derived Thbs4+GFAP⁺ astrocytes in the penumbra by 74% compared to control mice 6 days after PT (**Fig. 5c in the revised manuscript**).

Finally, we investigated the role of fibrinogen on the cell fate of SVZ-derived NSPCs by cell tracking using tamoxifen inducible *Nestin-CreER*^{T2};YFP^{fl} reporter transgenic mice. Remarkably, immunolabeling of SVZ-derived YFP⁺ cells in combination with Thbs4 and GFAP in the lesion area revealed a 64.5% reduction of SVZ-derived astrocytes in anicrod treated animals compared to control animals 10 days after PT, while the overall number of SVZ-derived YFP⁺Thbs4⁺ cells at the lesion scar was not changed (**Fig. 5d in the revised manuscript**), further substantiating our findings that fibrinogen-induced SVZ-generated Thbs4⁺ astrocytes migrated to the cortical lesion area and contributed to the glia scar after PT.

We performed additional, new experiments by elimination of fibrinogen using the *Fga*^{-/-} mouse line and BrdU cell tracking, which showed, in accordance to the anicrod data, a 50% reduction in the number of SVZ-derived BrdU⁺GFAP⁺ reactive astrocytes in the lesion area compared to control mice in the stab wound injury model (**Supplementary Fig. 9a-c**). Finally, we showed that fibrinogen was sufficient to induce SVZ adult NSPC differentiation into BrdU⁺GFAP⁺ astrocytes by stereotactically injecting fibrinogen into the cortex in close proximity to the SVZ (**Supplementary Fig. 9d-e**).

Overall, we added new data identifying Thbs4⁺ cells as a cellular source for the fibrinogen-induced increase in mature GFAP+S100 β ⁺ cells within the SVZ and showed that these cells significantly contribute to the cortical lesion site.

In addition, since gliogenesis often occurs at the expense of neurogenesis, it is striking that a study purporting to examine SVZ stem cell fate does not quantify olfactory bulb neurogenesis in vivo with a BrdU pulse labeling assay. In summary, while this is a potentially important work and the data flow logically, the approaches used limit the conclusions that can be drawn.

We thank the Reviewer for this constructive comment. Our new data suggest that fibrinogen deposition in the SVZ stem cell environment induces Thbs4⁺ proliferation and their maturation towards astrocytes after cortical brain injury and that these newly formed SVZ-derived astrocytes significantly contribute to the cortical lesion area. Deposition of fibrinogen in the SVZ after cortical injury has physiological relevance as this massively affects the distal cortical scar formation (change in pathology) by affecting SVZ NSPCs.

We performed additional experiments and investigated the role of fibrinogen on the neuronal cell fate of SVZ-derived NSPCs by cell tracking using tamoxifen inducible *Nestin-CreER^{T2};YFP^{fl}* reporter transgenic mice in fibrinogen-depleted mice 10 days after PT (**Supplementary Fig. 10 in the revised manuscript**).

Remarkably, immunolabeling of SVZ-derived YFP⁺ cells in combination with NeuN in the glomerular layer of the olfactory bulbs revealed a 2.6-fold increase in YFP⁺ cells and a 1.6-fold increase in YFP⁺NeuN⁺ cells in anicrod treated animals compared to control animals 10 days after PT (**Supplementary Fig. 10 in the revised manuscript**), further substantiating our findings that fibrinogen deposition within the SVZ after PT drives NSPC differentiation into astrocytes at the expense of olfactory bulb neurogenesis. We added new sentences to the result section on **page 13 in the revised manuscript**.

Fig 1: In Fig 1, they demonstrate GFAP reactivity and fibrinogen deposition in the SVZ after photothrombotic lesion, stab wound, and MCAO, as well as in human stroke. The authors argue based on localization that fibrinogen enters from blood vessels rather than the CSF, which seems reasonable but is difficult to prove. This language could be softened.

We agree that it is difficult to finally prove the route of entry for the deposition of fibrinogen in the SVZ stem cell niche, and thus we changed the sentence in the **revised manuscript** accordingly to “These results showed that a cortical brain insult leads to fibrinogen deposition into the distant SVZ stem cell niche, potentially by increased leakiness of the specialized SVZ vasculature.” on **page 6** of the result section.

Fig 4- The authors suggest that fibrinogen colocalizes and interacts with B1 integrin, an NSC marker, using a fractionated serum derived fibrinogen. Recombinant proteins lacking specific domains would certainly have been more elegant, as it is difficult to know what else might be different about this plasma fraction. Serum very potently drives astrogenesis and inhibits neurogenesis in vitro.

We thank the Reviewer for this constructive comment. We performed new experiments and further addressed, whether the fibrinogen-integrin interaction was necessary for the induction of BMP type I receptor activation and NSPC differentiation into astrocytes.

- We repeated our experiments with different fibrinogen isolate batches (**Fig. 4c-d in the revised manuscript**) to rule out potential fibrinogen isolate batch deviations.
- We added further information about the fibrinogen isolation procedure to the **material and method section in the revised manuscript** to describe the purity of the preparation of full-length fibrinogen and fibrinogen lacking most of the alphaC chain (**pages 21 in the revised manuscript**). The new paragraph named “Fibrinogen preparations” in the material and method section **in the revised manuscript** further described that full-length fibrinogen isolation and that the thrombin-coagulable fraction lacking the intact α C region was prepared by limited plasmin digestion of full-length fibrinogen. The fibrinogen fractions used (commercially available full-length fibrinogen, and isolated full-length as well as α C fibrinogen fractions) have different biological functions despite similar methods of isolation.
- Fibrinogen fractions do not contain free BMP (data not shown) and fibrinogen is not a carrier of BMP to induce Smad phosphorylation and adult NSPC differentiation into astrocytes (**Fig. 4a, Supplementary 7d-e in the revised manuscript**).
- Fibrinogen colocalizes with activated integrin β 1, fibrinogen-mediated NSPC differentiation into astrocytes was not induced by using fibrinogen isolates lacking the integrin-binding RGD containing alphaC-terminus and inhibition of lipid raft formation prevented fibrinogen-induced BMP signaling (**Fig. 4c-e and Supplementary Fig. 7f in the revised manuscript**). We added new experiments to investigate whether the RGD integrin recognition site on the fibrinogen alphaC chain is involved in NSPC differentiation into astrocytes. Pretreating NSPCs with an integrin-blocking peptide significantly reduced the fibrinogen-mediated NSPC differentiation into astrocytes (**Fig. 4f in the revised manuscript**).
- The *in vitro* data on a role of fibrinogen on NSPC differentiation into astrocytes are fully supported by the genetic evidence in *Fga^{-/-}* mice, which express all serum proteins except fibrinogen and

show a 74% reduction of GFAP+S100 β + astrocytes in the SVZ at 3 days post-injury compared to control mice (**Supplementary Fig. 4e in the revised manuscript**).

Overall, the *in vitro* and *in vivo* experiments support the role for fibrinogen on NSPC differentiation into astrocytes. Our study suggests the working model that fibrinogen via its integrin interacting α C domain induces integrin-dependent BMP type I receptor localization to lipid rafts to activate BMP signaling (Figure 6). The point of the Reviewer is well-taken and we softened our language in the revised manuscript and we added thoughts in the Discussion that future studies will be required to fully characterize the mechanism of fibrinogen-induced BMPR activation.

Fig 5 shows upregulation of biglycan protein after fibrinogen, although the significance of this is unclear, as it is just another astrocyte marker, thus not necessarily surprising.

We moved the biglycan data to the **Supplementary Fig. 6 of the revised manuscript** and did not remove the data to emphasize that fibrinogen-induced newborn astrocytes are not merely activated astrocytes, but rather a unique cell type secreting neuroprotective factors. Our new experiments showed that fibrinogen promotes Thbs4+ NSPC proliferation and their differentiation into mature astrocytes. These fibrinogen-driven SVZ-derived Thbs4+ astrocytes in the lesion area might have a beneficial role in brain repair via the Thbs4-Notch axis, as previously shown (Benner et al., 2013) and via fibrinogen-induced biglycan expression and secretion (**our study, Supplementary Fig. 6b**), a factor that has been shown to promote neuronal survival (Koops et al., 1996). We added sentences to **page 10 of the revised manuscript**, suggesting “different functionalities of fibrinogen-induced NSPC-derived newborn astrocytes compared to cortical resident astrocytes in the lesion area”.

Fig 5 also shows a pro astrogenic effects of fibrinogen on hIPSCs- quantification in 5c does not show the most relevant statistical result, which is control vs fibrinogen in the presence of CNTF

In agreement with the Reviewers (please see Reviewer 1 above) and to better focus the manuscript on the role of fibrinogen – BMP signaling in the SVZ and its contribution to brain repair we removed the human iPSC-derived progenitor cell aspects (**Fig. 5 of the original manuscript**), as these experiments are not necessary to understand the role of fibrinogen on SVZ NSPC differentiation into astrocytes.

Reviewer #3: In this manuscript the authors continue their studies on the role of fibrinogen in the generation and activation of astrocytes in the CNS. Specifically, the current manuscript investigates the effects of fibrinogen deposition in the subventricular zone following induction of a stroke or stab wound model. The primary model is a photothrombotic ischemia that induces significant unilateral damage to the CNS. The authors demonstrate that rapidly following stroke there is an increase in fibrinogen in the SVZ and this is correlated with an up-regulation of markers of mature astrocytes (GFAP etc.). The authors propose this is a consequence of induction of astrocyte fate in neural stem cells in the SVZ. To confirm their hypothesis the authors extend the study to an in vitro culture model and define the signaling pathways as being mediated through activation of the BMP pathway. Similar treatment of human iPS derived NSC resulted in a similar outcome in terms of the elevation of the number of mature astrocytes, while microarray analysis suggested the astrocytes that were generated by this paradigm had neuroprotective characteristics. Understanding the sequelae of CNS injury of neural cell fate is an important and timely issue. This manuscript clearly demonstrates that following a CNS insult there is a dramatic increase in the number of GFAP+ cells adjacent to germinal zones and at least part of that increase depends on the presence of fibrinogen. The manuscript in its current form does not, however provide compelling evidence that the increase in GFAP+ cells reflect fate changes in NSCs. Furthermore, the biological significance of the observations is missing. While the authors might suggest that induction of astrocytes enhances or inhibits CNS recovery this is not shown and it is equally possible that there is no change in pathology. In this case, while the biology may be interesting the impact of this manuscript is limited.

We thank the Reviewer for his insightful comments and suggestions. We performed additional experiments to address the two major points of the Reviewer, namely 1) to characterize how fibrinogen alters the cell

fate of NSPCs and to investigate the biological significance after cortical injury, and 2) to investigate how fibrinogen alters the lesion pathology.

To further strengthen that fibrinogen alters the cell fate choice of SVZ NSPCs and that fibrinogen-induced SVZ-derived newborn astrocytes contribute to the lesion area, we performed several new experiments:

We analyzed in detail the cell responses of the NSPC population in the SVZ at different timepoints after PT and found that fibrinogen induces the proliferation of the SVZ Thbs4⁺ NSPCs and their total number on day 3 after PT (**Fig. 2g in the revised manuscript, Supplementary Fig. 4d**). This Thbs4⁺ cells then further differentiate into S100 β +GFAP⁺ mature astrocytes within the SVZ (**Fig. 2h in the revised manuscript**). Importantly, fibrinogen depletion by anicrod or by using the fibrinogen-deficient mouse line drastically reduced the number of S100 β +GFAP⁺ mature astrocytes within the SVZ (**Fig. 2i and Supplementary Fig. 4e in the revised manuscript**). Interestingly, we could neither identify mature astrocyte proliferation in the first days after PT nor reactive astrocytes expressing activation marker, such as Lcn2, Serpina3n or CSPGs within the SVZ (**Supplementary Fig. 4b-c in the revised manuscript and data not shown**).

Next, we investigated whether fibrinogen-induced SVZ-generated Thbs4⁺ astrocytes migrated to the cortical lesion area and contributed to the glia scar after PT. We provide striking new data that fibrinogen depletion by anicrod drastically reduced the GFAP and CSPG expression level in the penumbra by 75% and 72%, respectively (**Fig. 5a-b in the revised manuscript**), revealing that fibrinogen alters the lesion pathology. Astrocytes adjacent to the vasculature proliferate after CNS injury and contribute to glial scar formation (Bardehle *et al.* 2013). However, depletion of fibrinogen did not affect local, perivascular astrocyte proliferation at the lesion site (**Supplementary Fig. 8**). Strikingly, fibrinogen depletion drastically reduced the number of SVZ-derived Thbs4⁺GFAP⁺ astrocytes in the penumbra by 74% compared to control mice 6 days after PT (**Fig. 5c in the revised manuscript**).

Finally, we investigated the role of fibrinogen on the cell fate of SVZ-derived NSPCs by cell tracking using tamoxifen inducible *Nestin-CreER^{T2};YFP^{fl}* reporter transgenic mice. Remarkably, immunolabeling of SVZ-derived YFP⁺ cells in combination with Thbs4 and GFAP in the lesion area revealed a 64.5% reduction of SVZ-derived astrocytes in anicrod treated animals compared to control animals 10 days after PT (**Fig. 5d in the revised manuscript**), while the overall number of SVZ-derived YFP⁺Thbs4⁺ cells at the lesion scar was not changed, further substantiating our findings that fibrinogen-induced SVZ-generated Thbs4⁺ astrocytes migrated to the cortical lesion area and contributed to the glia scar after PT.

We performed additional experiments by elimination of fibrinogen using the *Fga^{-/-}* mouse line and by cell tracking using BrdU labeling. These showed, in accordance to the anicrod data, a 50% reduction in the number of SVZ-derived BrdU⁺GFAP⁺ reactive astrocytes in the lesion area compared to control mice in the stab wound injury model (**Supplementary Fig. 9a-c**). Finally, we showed that fibrinogen was sufficient to induce SVZ-derived BrdU⁺ adult NSPC differentiation into BrdU⁺GFAP⁺ astrocytes by stereotactically injecting fibrinogen into the cortex in close proximity to the SVZ (**Supplementary Fig. 9d-e**).

Overall, these new data bolster further our finding that fibrinogen in the SVZ stem cell niche induces Thbs4⁺ NSPC proliferation and differentiation and that these newly formed SVZ-derived astrocytes significantly contribute to the cortical lesion area. Deposition of fibrinogen in the SVZ after cortical injury has physiological relevance as this massively affects the distal cortical scar formation (change in pathology) by affecting SVZ NSPCs.

Some additional issues that need to be addressed.

1. The issue of induced cell fate. The paper is very weak on descriptions of NSC. The quoted manuscript (Lugert *et al*) used a transgenic animal and FACs sorting – the current manuscript uses wild type animals.

We thank the Reviewer for his comment on the “induced cell fate” and the “description of the NSCs”. To address the question of “induced cell fate”, we FACS-isolated SVZ-derived NSPCs from

Nestin^{CreERT2}:Rosa26^{YFP} mice for *in vitro* experiments to conduct fate-mapping studies. The FACS-isolated SVZ-derived NSPCs were cultured directly under differentiation conditions. However, as shown in **Figure R2** below, although GFAP staining was evident and although fibrinogen treatment increased GFAP level, the cell morphology did not resemble astrocytes. We therefore conclude that the differentiation assay does not function properly using FACS-isolated cells.

Figure R2. Representative GFAP+ FACS-sorted untreated and fibrinogen-treated adult SVZ NSPCs (green) after 4 days on poly-D-lysine. Quantification of the GFAP immunoreactivity in the DRAQ5+ cells. 20 cells per condition were quantified. Scale bar, 110 μ m.

In addition, to further improve this differentiation assay, we cultured cells under differentiation conditions for different times and we plated cells on poly-D-lysine or laminin. However, the cells did not differentiate into proper astrocytes (higher GFAP expression + morphology).

In conclusion, we believe that our *in vitro* differentiation assay of SVZ-derived NSPCs derived from WT mice used in this manuscript is the superior system to directly investigate the role of fibrinogen on NSPC differentiation into astrocytes.

To address the question of the “description of the NSCs”, we performed new experiments and added more information to the revised manuscript. Wildtype SVZ- and hippocampal-derived NSPCs were isolated and used in our original manuscript. SVZ-derived NSPCs were isolated to generate neurospheres as described in the original paper by Reynolds *et al* in 1992 for the generation of the non-adherent neurosphere culture. These adult SVZ-derived NSPCs are multipotent stem cells in culture and are capable to differentiate into neurons, astrocytes and oligodendrocytes upon growth factor withdrawal and adherent culture conditions (Reynolds *et al.*, 1992). We performed new experiments and added new results to the revised manuscript revealing that the SVZ-derived NSPCs immediately after the initiation of differentiation (4 hours adherence and growth factor withdrawal, our timepoint of fibrinogen treatment) expressed typical stem cell marker (~95% SOX2 cells, ~95% Nestin cells, ~50% Ki-67 cells), but very few cells were positive for the glial precursor/astrocyte marker GFAP (~8.5%) or oligodendrocyte precursor cell marker NG2 (\leq 1%) (**Supplementary Fig. 3b and data not shown in the revised manuscript**). In addition, we added informations to the result section (**page 7 in the revised manuscript**) and to the material and method section and added new references (**page 23 in the revised manuscript**) to describe the hippocampal-derived NSPCs isolation and for the generation of the adherent culture of hippocampal NSPCs.

Overall, our new data confirm that we work with established multipotent NSPC cultures and that we are not working with astrocyte and/or oligodendrocyte precursor cells and that fibrinogen indeed drives the cell fate of the different SVZ-derived and hippocampal-derived stem cells towards astrocytes.

2. The characterization of the NSCs is missing. This is critical since the data may also be explained by the maturation of existing astrocyte/oligodendrocyte precursors (with no true fate switch). Indeed, this group has demonstrated such an effect in their 2017 neuron paper.

We thank the reviewer for his insightful comment to give a better description of NSPCs *in vitro* and *in vivo*. As mentioned above, we performed additional new experiments and added new results to the revised manuscript revealing that we work with established multipotent NSPC cultures and that we are not working with astrocyte and/or oligodendrocyte precursor cells and that fibrinogen indeed drives the cell fate of the different SVZ-derived and hippocampal-derived stem cells towards astrocytes.

Next, we described the SVZ NSPC population in detail after PT in the revised manuscript to further strengthen that fibrinogen indeed alters the cell fate choice of SVZ NSPCs:

Our data in **Fig. 2 of the original manuscript** showed that ischemic stroke induced fibrinogen deposition in the SVZ, which correlated with increased numbers of GFAP+ cells. Furthermore, we had used a marker combination for mature astrocytes that have lost their neural stem cell potential (GFAP+S100β+ cells, Raponi *et al.*, 2007) and we observed a significant increase in GFAP+S100β+ cells in the SVZ after ischemic stroke.

In our new experiments, we first investigated whether mature astrocyte proliferation or neural stem proliferation and differentiation contributed to this increased number of astrocytes after ischemic stroke. Detailed analysis of the SVZ subpopulation revealed that indeed the Thbs4+ NSPCs significantly increased until day 3 after PT. Interestingly, we detected a 5.7-fold increase in proliferating EdU+Thbs4+GFAP+ SVZ NSPCs 3 days after PT (**Fig. 2g and Supplementary Fig. 4d in the revised manuscript**) and immunolabeling of the Thbs4+ subpopulation in combination with GFAP and S100β revealed that the SVZ Thbs4+ NSPC subpopulation further matured into astrocytes (Thbs4+GFAP+S100β+; **Fig. 2h in the revised manuscript**). Importantly, fibrinogen depletion by anicrod and using the fibrinogen deficient mouse line (*Fga*^{-/-} mouse line) drastically reduced the number of S100β+GFAP+ mature astrocytes within the SVZ, respectively (**Fig. 2i and Supplementary Fig. 4e in the revised manuscript**).

The already mature resident SVZ GFAP+S100β+ astrocytes did not proliferate during the first days after PT, as shown by absent EdU labeling, suggesting that these cells are not the source for the increased number of astrocytes in the SVZ after PT (**Supplementary Fig. 4c in the revised manuscript**). Interestingly, we could not identify reactive astrocytes expressing activation marker, such as Lcn2, Serpina3n or CSPGs within the SVZ (**Supplementary Fig. 4b in the revised manuscript and data not shown**), suggesting that fibrinogen-induced NSPC differentiation into astrocytes within the SVZ is responsible for the increased number of GFAP+S100β+ cells after PT.

Overall, these new data further bolster our finding that in the SVZ stem cell niche fibrinogen induces Thbs4+ NSPC proliferation and differentiation and that these newly formed SVZ-derived astrocytes significantly contribute to the cortical lesion area.

3. The data in Fig 2 is difficult to interpret without data showing total cell number. Likewise, there is no attempt to examine cell proliferation or cell death either in vivo (Fig 2F) or in vitro.

We are grateful to the Reviewer for his insightful and constructive comments and we addressed the total cell number, apoptosis and proliferation of NSPCs *in vivo* and *in vitro*.

As detailed above, our new experiments analyzing total cell numbers, proliferation and apoptosis *in vivo* showed that fibrinogen induces Thbs4+ NSPC proliferation and differentiation and that these newly formed SVZ-derived astrocytes significantly contribute to the cortical lesion area. The total cell number and the number of apoptotic cells did not change significantly at any of the investigated timepoints in the SVZ after PT (**Supplementary Fig. 4d in the revised manuscript**).

Furthermore, fibrinogen treatment of SVZ NSPCs *in vitro* increased the cell number (**Supplementary Fig. 3f, top in the revised manuscript**) and decreased apoptosis (**Supplementary Fig. 3g, bottom in the revised manuscript**), while fibrinogen treatment did not alter hippocampal-derived NSPC cell numbers (**Supplementary Fig. 3f, bottom in the revised manuscript**).

Overall, these new data suggest that fibrinogen deposition in the SVZ stem cell environment induces Thbs4+ proliferation and their maturation towards astrocytes after cortical brain injury.

4. The human iPSC data seems very superficial and adds little to the overall story.

As mentioned above (**Reviewer 1**), we agree to remove the human iPSC-derived progenitor cell aspects (**Fig. 5 of the original manuscript**).

References

- Arvidsson A. *et al.* Neuronal replacement from endogenous precursors in the adult brain after stroke. *Nat Med* 8, 963-970 (2002).
- Bardehle S. *et al.* Live imaging of astrocyte responses to acute injury reveals selective juxtavascular proliferation. *Nat Neurosci* 16, 580-586 (2013).
- Benner E. J. *et al.* Protective astrogenesis from the SVZ niche after injury is controlled by Notch modulator Thbs4. *Nature* 497, 369-373 (2013).
- Bohrer C. *et al.* The balance of Id3 and E47 determines neural stem/precursor cell differentiation into astrocytes. *EMBO J* 34, 2804-2819 (2015).
- Faiz M. *et al.* Adult Neural Stem Cells from the Subventricular Zone Give Rise to Reactive Astrocytes in the Cortex after Stroke. *Cell Stem Cell* 17, 624-634 (2015).
- Hackett A. Y. *et al.* Injury type-dependent differentiation of NG2 glia into heterogeneous astrocytes. *Experimental Neurology* 308, 72-79 (2018).
- Koops A. *et al.* Cultured astrocytes express biglycan, a chondroitin/dermatan sulfate proteoglycan supporting the survival of neocortical neurons. *Brain Res Mol Brain Res* 41, 65-73 (1996).
- Lagace D. C. *et al.* Dynamic contribution of nestin-expressing stem cells to adult neurogenesis. *J Neurosci* 27, 12623-12629 (2007).
- Raponi E. *et al.* S100B expression defines a state in which GFAP-expressing cells lose their neural stem cell potential and acquire a more mature developmental stage. *Glia* 55, 165-177 (2007).
- Reynolds B. A. & Weiss S. Generation of neurons and astrocytes from isolated cells of the adult mammalian central nervous system. *Science* 255, 1707-1710 (1992).
- Schachtrup C. *et al.* Fibrinogen triggers astrocyte scar formation by promoting the availability of active TGF-beta after vascular damage. *J Neurosci* 30, 5843-5854 (2010).
- Zamanian J. L. *et al.*, Genomic analysis of reactive astrogliosis. *J Neurosci* 32, 6391-410 (2012).

Reviewers' comments:

Reviewer #1 (Remarks to the Author):

The authors have done a good job in responding to my comments. The paper is significantly improved with the addition of the SVZ-lineage tracing experiments and the emphasis on Thbs4 as a bona fide marker of SVZ derived reactive astrocytes, following the studies from Benner, et al. Solid revision.

Nevertheless, a few questions remain unanswered. Namely, the issue of local proliferation of astrocytes after PT. The Ki-67 staining is not sufficient to show this, as BrdU labeling is the preferred method. This is critical, because they report drastic effects on the generation of reactive astrocytes when they inhibit Fibrinogen and its highly unlikely that the entire phenomenon they are reporting is due to SVZ-derived reactive astrocytes. Thus, investigating whether local proliferation is impacted is very important. They claim, based on Ki-67 that its unaffected. Given the widespread effects of their drug and the fact that most reactive astrocytes are produced locally (not from the SVZ), I find their claims hard to believe.

Reviewer #2 (Remarks to the Author):

The authors have carefully strengthened their data and added supporting lines of evidence to address my major concerns. I feel that the manuscript is now suitable for publication.

I would only add a gentle reminder to the authors of the rebuttal not to assume that all reviewers are male. All three responses started with "we thank the reviewer for his insightful comments."

Reviewer #3 (Remarks to the Author):

In this revised manuscript the authors have made significant changes to the original paper and the story is considerably stronger as a result. The majority of the concerns of the original reviewers have been addressed in this revision, however there are still a couple of issues that would benefit from additional thought.

The authors have been very careful to present their data in terms of changes in astrocytes in the vicinity of the lesion and the contribution to glial scars. What they have not addressed is whether these changes have any effect on the outcome of the insult. While functional studies are beyond the scope of the current manuscript it would be interesting to know if there were changes in neuronal populations in the lesion core in the presence and absence of fibrinogen signaling.

A second is raised by the data in Figure 5. In this figure (5b) the Ancrod treated animal not only appears to lack fibrinogen but also totally lacks GFAP immunoreactivity. While reactive astrocytes are anticipated to be reduced by this treatment – this should not affect all GFAP+ cells. Indeed, in the Fga^{-/-} animals shown in figure S9 there is significant GFAP immunoreactivity. This raises the possibility that there are other "off target" effects of Ancrod that might complicate the interpretation of the data.

Point-by-Point Reply

We thank the editors and all three reviewers for their positive and constructive comments and for inviting us to resubmit a revised manuscript entitled “**Fibrinogen induces neural stem cell differentiation into astrocytes in the subventricular zone stem cell niche via BMP receptor signaling**”.

Prompted by the concern raised by the editors and Reviewer 1, we performed an additional experiment showing that fibrinogen does not significantly alter the proliferation nor the number of newly generated local astrocytes by using nucleoside analogue (EdU) labelling (**Supplementary Fig. 8b in the revised manuscript**). Furthermore, prompted by the concerns raised by Reviewers 1 and 3, we added sentences to page 12 in the result section and to pages 15-16 and pages 17-18 in the discussion section in the revised manuscript to have a more thorough and nuanced description of the results and the discussion of our results.

Overall, with the new experiment and the new information provided, we strongly believe that all the conclusions of the paper are adequately supported and that we have bolstered our original findings on the novel mechanism that fibrinogen drives NSPC differentiation into astrocytes within the SVZ and that these newly generated SVZ-derived astrocytes contribute to lesion scar formation. A detailed point-by-point reply follows below and all changes in the revised manuscript are underlined.

Reviewer #1 (Remarks to the Author):

The authors have done a good job in responding to my comments. The paper is significantly improved with the addition of the SVZ-lineage tracing experiments and the emphasis on Thbs4 as a bona fide marker of SVZ derived reactive astrocytes, following the studies from Benner, et al. Solid revision. Nevertheless, a few questions remain unanswered. Namely, the issue of local proliferation of astrocytes after PT. The Ki-67 staining is not sufficient to show this, as BrdU labeling is the preferred method. This is critical, because they report drastic effects on the generation of reactive astrocytes when they inhibit Fibrinogen and its highly unlikely that the entire phenomenon they are reporting is due to SVZ-derived reactive astrocytes. Thus, investigating whether local proliferation is impacted is very important. They claim, based on Ki-67 that its unaffected. Given the widespread effects of their drug and the fact that most reactive astrocytes are produced locally (not from the SVZ), I find their claims hard to believe.

As suggested, we performed the EdU labelling experiment to address local proliferation of astrocytes. Our new data showed that depletion of fibrinogen did not affect local, perivascular astrocyte proliferation and the formation of new cells at the lesion site (**new Supplementary Fig. 8b in the revised manuscript**).

The dramatic effect of fibrinogen depletion on GFAP and CSPG immunoreactivity in the scar (Fig. 5b) includes effects on resident cortical astrocytes (which we had shown before to be activated by fibrinogen via activation of the TGFbeta receptor signaling pathway (Schachtrup et al. 2010)) and on SVZ-derived newborn astrocytes (this study). To avoid the impression that fibrinogen-induced SVZ-derived new-born astrocytes are solely responsible for the reactive astrocyte scar in the lesion area, we added further information to the result section in the revised manuscript on **page 12** stating that “*Fibrinogen depletion by anocrod reduced GFAP and CSPG expression levels in the penumbra by 75% and 72%, respectively, confirming a robust effect of fibrinogen on the activation status of the astrocyte population forming the glial scar in the lesion area (Schachtrup et al., 2010), including local and SVZ-derived newborn astrocytes (Wanner et al., 2013, Bardehle et al., 2013, Benner et al., 2013, Bohrer et al., 2015, Faiz et al., 2015) (Fig. 5a-b).*”. In the discussion, we added sentences to **pages 15-16** stating that “*Future studies will show whether Fibrinogen-induced SVZ-derived Thbs4+GFAP+ newborn astrocytes that reach the cortical lesion area will become responsive to fibrinogen-bound TGF-β and whether these cells contribute to the inhibitory scar or if these newborn astrocytes will have other functions in the lesion area with a more beneficial role in brain repair.*” and to pages 17-18 stating that “*Our results showed that a cortical brain insult leads to fibrinogen deposition into the SVZ and, as expected into the cortical lesion area (Fig. 1b). The cortical astrocyte scar in the lesion area after PT consists of resident and locally produced reactive astrocytes, as well as of SVZ-derived newborn astrocytes (Wanner et al., 2013, Bardehle et al., 2013, Benner et al., 2013, Bohrer et al., 2015, Faiz et al., 2015). Fibrinogen provokes astrocyte differentiation from NSPCs via BMP receptor signaling (this study) as well as astrocyte activation by promoting the availability of active TGF-beta after vascular damage (Schachtrup et al., 2010). Astrocytes adjacent to the vasculature proliferate after CNS injury and are a major source to glial scar formation. However, depletion of fibrinogen did not affect the proliferation and formation of new (Ki67+/EdU+) local, perivascular astrocytes at the lesion site*

(Supplementary Fig. 8), suggesting that in the lesion area fibrinogen deposition does not affect overall astrocyte cell number, but its activation status.“).

Reviewer #2: The authors have carefully strengthened their data and added supporting lines of evidence to address my major concerns. I feel that the manuscript is now suitable for publication. I would only add a gentle reminder to the authors of the rebuttal not to assume that all reviewers are male. All three responses started with "we thank the reviewer for his insightful comments."

We are grateful to the Reviewer for appreciating the new data included during the revision and considering our manuscript suitable for publication. We apologize for wrong usage of the gender article. This was not on purpose and we will pay more attention to this in the future.

Reviewer #3: In this revised manuscript the authors have made significant changes to the original paper and the story is considerably stronger as a result. The majority of the concerns of the original reviewers have been addressed in this revision, however there are still a couple of issues that would benefit from additional thought. The authors have been very careful to present their data in terms of changes in astrocytes in the vicinity of the lesion and the contribution to glial scars. What they have not addressed is whether these changes have any effect on the outcome of the insult. While functional studies are beyond the scope of the current manuscript it would be interesting to know if there were changes in neuronal populations in the lesion core in the presence and absence of fibrinogen signaling.

We completely agree with the Reviewer that it will be interesting to address the role of fibrinogen on cortical neuron survival/regeneration and its impact on functional outcome. However, the current manuscript focuses on the role of fibrinogen in NSPC differentiation into astrocytes in the SVZ stem cell niche and thus investigating neuronal populations in the cortical lesion area will be our focus in future studies.

A second is raised by the data in Figure 5. In this figure (5b) the Ancrod treated animal not only appears to lack fibrinogen but also totally lacks GFAP immunoreactivity. While reactive astrocytes are anticipated to be reduced by this treatment – this should not affect all GFAP+ cells. Indeed, in the Fga-/- animals shown in figure S9 there is significant GFAP immunoreactivity. This raises the possibility that there are other “off target” effects of Ancrod that might complicate the interpretation of the data.

The reviewer raises an intriguing question, why GFAP immunoreactivity is differentially affected by acute fibrinogen depletion compared to the effects in fibrinogen deficient mice. We suggest that genetic deletion of fibrinogen *visa* acute fibrinogen depletion differently affects proteins of the coagulation cascade, clot formation and the formation of the provisional matrix in the lesion area, which has not been described and studied, yet. Acute fibrinogen depletion results in a small percentage of remaining circulating fibrinogen that is just enough for immediate hemostasis function after an injury, while fibrinogen knockout mice have an altered hemostasis because of the lack of fibrinogen. Thus, rather than relevant “off-target effects” of ancrod, we think that compensatory mechanisms in the *Fga*^{-/-} mice are responsible for the slight different phenotypes, however overall we confirmed our basic claims using either acute depletion, complete KO or induction of fibrinogen.

Reference

- Bardehle, S. *et al.* Live imaging of astrocyte responses to acute injury reveals selective juxtavascular proliferation. *Nat Neurosci* **16**, 580-586 (2013).
- Benner, E. J. *et al.* Protective astrogenesis from the SVZ niche after injury is controlled by Notch modulator Thbs4. *Nature* **497**, 369-373 (2013).
- Bohrer, C. *et al.* The balance of Id3 and E47 determines neural stem/precursor cell differentiation into astrocytes. *EMBO J* **34**, 2804-2819, doi:10.15252/embj.201591118 (2015).
- Faiz, M. *et al.* Adult Neural Stem Cells from the Subventricular Zone Give Rise to Reactive Astrocytes in the Cortex after Stroke. *Cell Stem Cell* **17**, 624-634, doi:10.1016/j.stem.2015.08.002 (2015).
- Schachtrup C. *et al.* Fibrinogen triggers astrocyte scar formation by promoting the availability of active TGF-beta after vascular damage. *J Neurosci* **30**, 5843-5854 (2010).
- Wanner, I. B. *et al.* Glial scar borders are formed by newly proliferated, elongated astrocytes that interact to corral inflammatory and fibrotic cells via STAT3-dependent mechanisms after spinal cord injury. *J Neurosci* **33**, 12870-12886 (2013).

REVIEWERS' COMMENTS:

Reviewer #1 (Remarks to the Author):

The authors have adequately addressed my concerns.